# Geraniin Mitigates Neuropathic Pain Through Antioxidant, Anti-Inflammatory, and Nitric Oxide Modulation in a Rat Model of Chronic Constriction Injury

**DOI:** 10.3390/ijms27010507

**Published:** 2026-01-03

**Authors:** Chih-Chuan Yang, Mao-Hsien Wang, Yi-Wen Lin, Chih-Hsiang Fang, Yu-Chuan Lin, Kuo-Chi Chang, Cheng-Chia Tsai

**Affiliations:** 1Department of Neurosurgery, Mackay Memorial Hospital, Taipei 10449, Taiwan; 2Department of Nursing, Mackay Junior College of Medicine, Nursing and Management, Taipei 11260, Taiwan; 3Department of Anesthesia, En Chu Kon Hospital, New Taipei City 23702, Taiwan; 4Institute of Biomedical Engineering, National Taiwan University, Taipei 10051, Taiwan; 5Department of Chinese Medicine, China Medical University Hospital, Taipei 114, Taiwan; 6National Applied Research Laboratories, Institute of Taiwan Instrument Research, Hsinchu 300092, Taiwan; 7Department of Chemical Engineering and Biotechnology, National Taipei University of Technology, Taipei 10608, Taiwan; 8Department of Medicine, Mackay Medical College, New Taipei City 252, Taiwan

**Keywords:** chronic constriction injury, geraniin, inflammation, neuropathic pain, nitric oxide

## Abstract

Neuropathic pain (NPP) remains therapeutically challenging, with oxidative/nitrosative stress and neuroinflammation—amplified by nitric oxide (NO)—as key drivers. This study investigated geraniin (GRN), a naturally occurring hydrolyzable ellagitannin widely distributed in various plant species, including *Phyllanthus* spp. and *Nephelium lappaceum* (rambutan), in a rat model of sciatic nerve chronic constriction injury (CCI), focusing on NO-pathway involvement. Male Wistar rats (n = 8/group) received intraperitoneal GRN (3, 10, 30, or 100 mg/kg) or vehicle (1% DMSO in saline) daily for 21 days. Behavioral (thermal hyperalgesia, mechanical allodynia, sciatic functional index), electrophysiological (nerve conduction velocity), and biochemical markers—oxidative/nitrosative stress (nitrite, MDA), antioxidant defenses (GSH, SOD, CAT), inflammation (TNF-α, IL-1β, IL-6, MPO), and apoptosis (caspase-3)—were quantified. L-arginine or L-NAME was co-administered to probe NO signaling. GRN at 30 and 100 mg/kg produced significant antinociceptive and neuroprotective effects; 30 mg/kg was selected for detailed analysis. By day 21, GRN improved pain thresholds and nerve conduction, enhanced antioxidant capacity, suppressed inflammatory mediators, and reduced caspase-3 activity. L-arginine reversed, whereas L-NAME potentiated these effects, confirming NO-dependent modulation. Collectively, GRN mitigates CCI-induced NPP via coordinated antioxidant, anti-inflammatory, and anti-apoptotic actions, supporting its potential as a multi-target candidate for pharmacokinetic and translational development.

## 1. Introduction

Neuropathic pain (NPP) is defined by the International Association for the Study of Pain (IASP) as pain arising from a lesion or disease of the somatosensory nervous system [1,2]. It manifests as spontaneous pain, hyperalgesia, allodynia, dysesthesia, and sensory loss—symptoms reflecting maladaptive changes in both peripheral and central neural circuits. Affecting approximately 7–18% of the global population, NPP is a chronic and debilitating condition that profoundly impairs quality of life and imposes substantial socioeconomic and healthcare burdens [1,2]. Current pharmacotherapies—including nonsteroidal anti-inflammatory drugs, opioids, anticonvulsants, antidepressants, and topical agents—often provide only partial and transient relief while producing dose-limiting side effects [2,3]. The limited efficacy of existing treatments underscores the urgent need for novel therapeutic strategies with multitarget mechanisms and improved safety profiles.

Pathophysiologically, NPP persists through peripheral and central sensitization involving aberrant nociceptor excitability, dysregulated ion channel and receptor expression, altered synaptic plasticity, and transcriptional reprogramming within dorsal root ganglia [3,4]. Mounting evidence implicates oxidative and nitrosative stress, together with neuroinflammatory signaling, as major contributors to this process [5,6]. Among these, nitric oxide (NO) serves as a key mediator that amplifies neuroinflammation, induces nitrosative damage, and enhances neuronal excitability [7,8,9,10]. Accordingly, inhibition of NO signaling, along with reinforcement of antioxidant and anti-inflammatory defenses, has been shown to attenuate neuropathic hypersensitivity in preclinical models [5,6,7,10,11,12].

Natural compounds with pleiotropic pharmacological actions have attracted increasing interest as alternative or adjunctive agents for chronic pain management [13,14].

Geraniin (GRN) is a hydrolyzable ellagitannin widely distributed in several medicinal and edible plants, particularly in *Phyllanthus* species and in the peel of *Nephelium lappaceum* (rambutan). Previous studies have demonstrated that geraniin exhibits a broad range of biological activities, including antioxidant, anti-inflammatory, and cytoprotective effects [15,16,17]. GRN has demonstrated neuroprotective effects in diverse experimental paradigms, including cerebral ischemia/reperfusion, spinal cord injury, nephrotoxicity, and chemically induced neurotoxicity, primarily via free radical scavenging, activation of endogenous antioxidant systems, and suppression of pro-apoptotic and pro-inflammatory pathways [18,19]. However, its potential therapeutic value in peripheral neuropathic pain—particularly within the chronic constriction injury (CCI) model—remains largely unexplored.

The present study aimed to elucidate the antinociceptive and neuroprotective mechanisms of GRN in a rat model of CCI-induced NPP. Behavioral (thermal hyperalgesia, mechanical allodynia, sciatic functional index), electrophysiological (nerve conduction velocity), and biochemical assessments targeting oxidative/nitrosative stress, antioxidant defenses, inflammation, and apoptosis were performed. To further delineate the role of NO signaling, GRN was co-administered with L-arginine (a NO precursor) or L-NAME (a nonselective NOS inhibitor). Rigorous statistical analyses incorporating corrected degrees of freedom and transparent reporting of individual data points were implemented to ensure analytical robustness and reproducibility. Collectively, this work provides the first systematic characterization of GRN in CCI-induced NPP and identifies it as a promising multitarget candidate for future therapeutic development.

## 2. Results

During the 21-day treatment period, animals were weighed daily as part of routine health monitoring and for dose adjustment. No statistically significant differences (*p* > 0.05) in body weight trajectories were observed among the experimental groups (Figure 1), and no treatment-related weight loss or signs of compound-related toxicity were detected throughout the study.

### 2.1. GRN Reduces CCI-Evoked Thermal Hypersensitivity

Following chronic constriction injury (CCI), rats in the injury group (CI) exhibited a marked reduction in paw-withdrawal latency, confirming the development of persistent thermal hyperalgesia (Figure 2a and Figure 3a). Repeated-measures two-way ANOVA revealed a significant group × time interaction (F(32, 95) = 3.60, *p* < 0.001), indicating differential temporal responses among treatment groups.

Geraniin (GRN) treatment significantly attenuated CCI-induced thermal hypersensitivity in a dose- and time-dependent manner. Antinociceptive effects became evident from day 14 at doses ≥30 mg/kg and persisted through day 21. At the end of the study, GRN at 30 and 100 mg/kg produced comparable and pronounced prolongation of withdrawal latency, whereas the 10 mg/kg dose exerted a more modest effect, suggesting a plateau in the dose–response relationship (Figure 2b).

Pharmacological modulation supported the involvement of nitric oxide (NO) signaling. Co-administration of L-arginine abolished the analgesic effect of GRN, whereas L-NAME significantly potentiated it (Figure 3a,b).

### 2.2. GRN Reverses CCI-Induced Mechanical Allodynia

CCI induced a sustained decrease in paw-withdrawal threshold (PWT), consistent with mechanical allodynia (Figure 4a and Figure 5a). Repeated-measures two-way ANOVA demonstrated a significant group × time interaction (F(32, 95) = 3.18, *p* < 0.001).

GRN administration significantly increased PWT values at later time points, with improvement becoming apparent from day 14 at doses ≥30 mg/kg and persisting through day 21. The 30 and 100 mg/kg doses produced comparable antihyperalgesic effects, whereas the lowest dose was less effective (Figure 4b). L-arginine co-treatment abolished, while L-NAME enhanced, the protective effects of GRN (Figure 5a,b), indicating a critical role of NO signaling.

### 2.3. GRN Promotes Functional Recovery of the Sciatic Nerve

CCI caused a progressive deterioration of sciatic nerve function, reflected by a marked decline in the sciatic functional index (SFI) (Figure 6a and Figure 7a). Repeated-measures analysis showed a significant group × time interaction (F(32, 95) = 3.60, *p* < 0.001), indicating treatment-dependent differences across time.

GRN treatment improved SFI values during the subacute and chronic phases following injury. GRN at 30 mg/kg enhanced functional recovery from day 7 onward, with no additional benefit observed at 100 mg/kg, suggesting a ceiling effect. L-arginine negated, whereas L-NAME potentiated, GRN-induced functional recovery (Figure 7a,b).

### 2.4. GRN Improves Sciatic Nerve Conduction

Electrophysiological recordings at day 21 showed that CCI significantly reduced both motor and sensory nerve conduction velocities (MNCV and SNCV) compared with control animals (Figure 8a,b). One-way ANOVA revealed significant group differences for both parameters (MNCV: F(4, 35) = 9.12, *p* < 0.001; SNCV: F(4, 35) = 8.47, *p* < 0.001).

GRN treatment markedly restored both MNCV and SNCV, indicating improved functional integrity of the injured sciatic nerve. These improvements were attenuated by L-arginine and further enhanced by L-NAME, supporting the involvement of NO-dependent mechanisms.

### 2.5. GRN Dampens Oxidative/Nitrative Stress in Sciatic Nerves

CCI markedly elevated nitrite and malondialdehyde (MDA) levels in sciatic nerve tissue, reflecting increased nitrative and oxidative stress (Figure 9a,b). One-way ANOVA demonstrated significant group effects for both nitrite and MDA levels (nitrite: F(4, 35) = 18.67, *p* < 0.001; MDA: F(4, 35) = 22.94, *p* < 0.001).

GRN treatment significantly reduced both indices of redox stress. These antioxidant effects were abolished by L-arginine and further strengthened by L-NAME, indicating that modulation of NO signaling critically influences GRN-mediated redox regulation.

### 2.6. GRN Restores Endogenous Antioxidant Defenses

CCI resulted in substantial depletion of endogenous antioxidant defenses, including glutathione (GSH), superoxide dismutase (SOD), and catalase (CAT) activities (Figure 10a–c). One-way ANOVA revealed significant group differences for all antioxidant parameters (GSH: F(4, 35) = 17.53; SOD: F(4, 35) = 10.84; CAT: F(4, 35) = 14.76; all *p* < 0.001).

GRN administration significantly restored GSH levels and SOD and CAT activities toward control values. L-arginine prevented this restoration, whereas L-NAME further potentiated the antioxidant effects of GRN, indicating NO-dependent regulation of endogenous antioxidant capacity.

### 2.7. GRN Suppresses Neuroinflammatory Mediators

CCI robustly increased pro-inflammatory cytokines (TNF-α, IL-1β, and IL-6) and myeloperoxidase (MPO) activity in sciatic nerve tissue at day 21 (Figure 11a–d). One-way ANOVA demonstrated significant group effects for all inflammatory markers (TNF-α: F(4, 35) = 29.41; IL-1β: F(4, 35) = 31.76; IL-6: F(4, 35) = 16.28; MPO: F(4, 35) = 24.93; all *p* < 0.001).

GRN treatment markedly reduced cytokine levels and MPO activity, indicating a strong anti-inflammatory effect. These effects were abolished by L-arginine and further potentiated by L-NAME, highlighting the modulatory role of NO signaling in GRN-mediated suppression of neuroinflammation.

### 2.8. GRN Modulates Apoptosis-Related Protein Expression

CCI significantly increased caspase-3 activity in sciatic nerve tissue, indicating enhanced apoptosis-related signaling (Figure 12). One-way ANOVA revealed a significant group effect for caspase-3 activity (F(4, 35) = 21.62, *p* < 0.001).

GRN treatment markedly reduced caspase-3 activity, suggesting protection against apoptosis-associated nerve damage. This anti-apoptotic effect was negated by L-arginine and further potentiated by L-NAME, supporting the involvement of NO-dependent pathways in GRN-mediated neuroprotection.

## 3. Discussion

The present study demonstrates that geraniin (GRN), a naturally occurring hydrolyzable ellagitannin with established antioxidant and anti-inflammatory properties, effectively alleviates neuropathic pain–like behaviors and improves peripheral nerve function in rats subjected to chronic constriction injury (CCI) of the sciatic nerve. These beneficial effects were associated with alterations in nitric oxide (NO) signaling, partial restoration of redox homeostasis, and attenuation of neuroinflammatory and apoptotic markers. Importantly, while the general antioxidant activity of GRN has been reported previously, this study provides integrated functional, biochemical, and pharmacological evidence supporting its therapeutic relevance in an experimental model of neuropathic pain. Nevertheless, the findings should be interpreted cautiously in light of methodological and mechanistic limitations.

The CCI model reproduced hallmark features of peripheral neuropathy, including thermal hyperalgesia, mechanical allodynia, impaired sciatic functional index, and reduced motor and sensory nerve conduction velocities (MNCV/SNCV). These functional deficits, which are consistent with axonal degeneration and demyelination, were significantly ameliorated by GRN treatment. The concurrent improvement in behavioral outcomes and electrophysiological parameters suggests that GRN provides not only symptomatic pain relief but also partial preservation of peripheral nerve integrity.

At the biochemical level, CCI induced pronounced oxidative and nitrosative stress, as evidenced by elevated malondialdehyde (MDA) and nitrite levels together with depletion of endogenous antioxidant defenses (GSH, SOD, and CAT) [20]. GRN effectively reversed these alterations, consistent with its reported redox-modulating capacity [5,6]. Excessive NO production, primarily attributed to inducible nitric oxide synthase (iNOS), can react with superoxide to generate peroxynitrite, thereby amplifying oxidative damage. The observed reduction in nitrite levels therefore suggests decreased NO generation and/or enhanced scavenging of reactive nitrogen species. However, because the relative contributions of iNOS, nNOS, and eNOS were not directly assessed, isoform-specific mechanisms remain unresolved and warrant further investigation [21,22].

In parallel, GRN markedly reduced levels of TNF-α, IL-1β, IL-6, and myeloperoxidase (MPO) activity—key mediators that perpetuate nociceptive signaling and neuroinflammation [15,18,19]. GRN also attenuated caspase-3 activation, indicating protection against apoptosis-associated neuronal and glial loss. Collectively, these findings suggest that GRN disrupts a redox–inflammatory–apoptotic axis central to CCI pathology. Pharmacological modulation further supports this interpretation, as L-arginine attenuated whereas L-NAME potentiated the protective effects of GRN [23,24]. Nevertheless, because L-NAME is a nonselective nitric oxide synthase inhibitor, these results indicate an association with NO signaling rather than establishing direct causality. Future studies using isoform-selective NOS inhibitors or direct enzymatic and molecular assays are needed to delineate the precise NO-related targets of GRN.

From a comparative perspective, GRN belongs to a broader class of bioactive polyphenols and ellagitannins that have been investigated for neuroprotective and anti-inflammatory effects. Compounds such as punicalagin, another well-characterized ellagitannin, and flavonoids such as quercetin have been reported to attenuate neuropathic pain and neuroinflammation primarily through antioxidant, NF-κB–inhibitory, and Nrf2-activating mechanisms. Although the present study did not include a reference antioxidant control group, comparison with these related compounds suggests that GRN exhibits a comparable multitarget profile, with the additional feature of pronounced NO-related modulation. Inclusion of benchmark antioxidants or structurally related ellagitannins in future studies would be valuable for defining the relative efficacy and mechanistic specificity of GRN within this compound class.

Among the tested doses, 30 mg/kg produced maximal therapeutic benefit without additional improvement at 100 mg/kg, suggesting a pharmacodynamic ceiling that may be related to saturable absorption or metabolic conversion. As a hydrolyzable ellagitannin, GRN can undergo intestinal biotransformation into ellagic acid and subsequently urolithins via gut microbiota–mediated metabolism. These metabolites have been reported to exert intrinsic antioxidant and anti-inflammatory activities through modulation of key signaling pathways, including suppression of NF-κB and iNOS signaling, activation of the Nrf2/ARE antioxidant response, and regulation of MAPK cascades [25,26,27]. The neuroprotective effects observed in the present study are therefore likely mediated by multiple convergent molecular mechanisms rather than a single pathway. Comprehensive pharmacokinetic and metabolic studies are required to define GRN bioavailability, active metabolites, and therapeutic exposure ranges in vivo.

With respect to future perspectives and clinical translation, several considerations emerge. First, validation of GRN efficacy in additional neuropathic pain models—such as chemotherapy-induced, diabetic, or traumatic neuropathy—will be essential to establish broader translational relevance. Second, alternative routes of administration, including oral delivery or localized perineural application, should be explored to better approximate clinical scenarios. Third, given that current neuropathic pain management often relies on combination therapy, GRN may represent a promising adjunct to existing treatments (e.g., gabapentinoids or antidepressants), potentially allowing dose reduction and mitigation of adverse effects. Finally, the development of optimized formulations to enhance bioavailability and consistency of exposure will be critical steps toward clinical application.

Consistency with previous geraniin studies further strengthens the biological plausibility of these findings. GRN has been reported to exert significant anti-inflammatory, antioxidant, and cytoprotective effects in diverse in vivo models, with effective doses typically ranging from 10 to 100 mg/kg [15,18,19]. The efficacy observed in the present study (30 mg/kg, i.p., for 21 days) aligns well with these earlier investigations. However, most available studies, including the current work, were conducted exclusively in male rodents. Given accumulating evidence for sex-dependent differences in neuropathic pain mechanisms and drug responsiveness, future studies incorporating both sexes are essential to improve translational relevance.

Several limitations should be acknowledged. NOS isoforms were not directly assessed, precluding isoform-specific interpretation of NO involvement. Pharmacokinetic and metabolic data for GRN and its metabolites are currently lacking, preventing definitive dose–exposure correlations. In addition, histopathological confirmation of axonal and myelin integrity, as well as molecular assessment of Nrf2 activation and glial reactivity, would further strengthen mechanistic understanding and should be incorporated in future investigations.

## 4. Materials and Methods

### 4.1. Animals

Male Wistar rats (total: 82 rats; 12 weeks old; 300–320 g at study onset) were used in this study. Animals were acclimated for at least one week in a temperature- and humidity-controlled vivarium (22 ± 2 °C; 55 ± 10% relative humidity) under a 12 h light/dark cycle, with free access to standard laboratory chow and water ad libitum. No fasting period was imposed prior to behavioral or biochemical assessments. All experimental procedures conformed to the U.S. National Institutes of Health guidelines for the care and use of laboratory animals and the International Association for the Study of Pain (IASP) Guidelines for the Use of Laboratory Animals in Pain Research, and were approved by the Institutional Animal Care and Use Committee (IACUC) of the National Taiwan University College of Medicine and College of Public Health (Approval No. 20220525). Animals were monitored daily for general health status, body weight, and signs of distress. Humane endpoint criteria were predefined, including severe motor impairment, persistent self-mutilation, signs of systemic illness, or >20% body-weight loss [18,19]. Behavioral testing was conducted in a temperature-controlled testing room between 09:00 and 14:00. To reduce animal use, group sizes were kept to the minimum compatible with prior studies and expected effect sizes. At study completion, animals were euthanized by CO_2_ overdose in accordance with IACUC guidelines.

### 4.2. Drugs

Geraniin (GRN; ≥95.0% purity; CAS 60976-49-0; PHL80994, Sigma, St. Louis, MO, USA) was dissolved initially in 1% DMSO and brought to volume with sterile normal saline. L-arginine (LA; CAS 74-79-3; A8094-25G) and L-NAME (LN; CAS 51298-62-5; 483125-100MG) were obtained from Sigma and prepared in normal saline. All solutions were freshly prepared on dosing days. Drug solutions were freshly prepared before use. Drug dosages were adapted from previous publications [10,18] and administered intraperitoneally (i.p.) once daily for 21 days at a volume of 2.0 mL/kg body weight.

### 4.3. Induction of Peripheral Neuropathy (CCI of the Sciatic Nerve)

Peripheral neuropathy was induced using the chronic constriction injury (CCI) model of the sciatic nerve, as originally described by Bennett and Xie, with minor modifications [28]. Rats were anesthetized with chloral hydrate (300 mg/kg, i.p.) and maintained under aseptic conditions throughout the procedure. A lateral incision was made at the mid-thigh level to expose the left common sciatic nerve. Four loose ligatures (4-0 chromic gut sutures) were placed around the nerve with approximately 1 mm spacing, just tight enough to induce slight constriction without interrupting epineurial blood flow. The muscle and skin were then sutured in layers. Sham-operated animals underwent identical surgical exposure of the sciatic nerve without ligation. Following surgery, animals were allowed to recover on a warming pad and were returned to their home cages. Postoperative recovery was monitored daily, including assessment of wound condition, body weight, locomotor activity, and general health status. Development of neuropathic pain was verified by the presence of mechanical allodynia and thermal hyperalgesia, assessed using von Frey filament testing and thermal nociceptive assays, respectively. Only animals that exhibited stable neuropathic pain–like behaviors were included in subsequent experimental procedures.

### 4.4. Experimental Design and Treatment Allocation

The experimental design is summarized schematically in Figure 12 to provide an at-a-glance overview of group allocation, treatment regimens, and experimental timelines. The study consisted of two sequential phases: (i) a dose-finding phase to determine the effective dose range of geraniin (GRN), and (ii) a mechanistic phase to examine the involvement of nitric oxide (NO) signaling. Animals were weighed daily during the treatment period for routine health monitoring and dose adjustment, randomly assigned to experimental groups, and all behavioral, electrophysiological, and biochemical assessments were conducted by investigators blinded to treatment allocation.

Dose-Finding Phase

(Rats received daily intraperitoneal injections for 21 days starting on day 1; n = 8/group)

I. Control (CT): anesthesia only; vehicle (1% DMSO in saline).

II. Sham (SM): sciatic nerve exposure without ligation; vehicle administered.

III. CCI (CI): chronic constriction injury with vehicle administered.

IV–VII. CCI+GRN: CCI with GRN administered at 3, 10, 30, or 100 mg/kg (CI+GN3, CI+GN10, CI+GN30, CI+GN100).

Behavioral assessment at day 21 demonstrated robust antinociceptive efficacy at GRN doses of 30 and 100 mg/kg, with no significant difference between these two doses. Based on these results, 30 mg/kg was selected as the optimal dose for subsequent mechanistic studies.

Mechanistic Phase (NO Pathway Modulation)

(n = 8/group)

VIII. CCI+LA+GRN (CI+LA+GN30): CCI rats received L-arginine (LA, 100 mg/kg) followed by GRN (30 mg/kg).

IX. CCI+LN+GRN (CI+LN+GN30): CCI rats received L-NAME (LN, 10 mg/kg) followed by GRN (30 mg/kg).

For groups VIII and IX, LA or LN was administered 60 min prior to GRN injection.

Behavioral tests were conducted one day before surgery (day −1) and on days 3, 7, 14, and 21 following CCI, with assessments performed 60 min after drug administration on each testing day. Following the final behavioral evaluation on day 21, selected groups (CT, SM, CI, CI+GN30, CI+LA+GN30, and CI+LN+GN30) underwent electrophysiological recordings of motor and sensory nerve conduction velocities. Animals were euthanized approximately 1 h after electrophysiological assessment, and sciatic nerve tissues were collected immediately for biochemical analyses. Throughout the study, animals were monitored daily for general health and postoperative recovery. Any losses were recorded and were not associated with GRN treatment. A schematic representation of the experimental workflow, group allocation, dosing schedule, and assessment timeline is provided in Figure 13.

### 4.5. Behavioral Assessments

#### 4.5.1. Thermal Nociception (Hot-Plate)

Thermal hyperalgesia was assessed using a hot-plate apparatus (Socrel DS-35; Ugo Basile, Comerio, VA, Italy) maintained at 45 ± 0.2 °C. Rats were placed in an open-ended cylinder on the plate, and the latency to the first nocifensive behavior (licking the left hind paw or jumping) was recorded in seconds; a 180 s cut-off prevented tissue injury [29].

#### 4.5.2. Mechanical Sensitivity (Von Frey)

Mechanical allodynia was assessed using calibrated von Frey filaments (Bioseb, Chaville, France) according to the method described by Chaplan et al. (1994) [30]. Animals were placed individually in transparent acrylic enclosures on an elevated wire-mesh platform and allowed to acclimate for at least 60 min prior to testing. Behavioral testing was conducted in a temperature-controlled testing room (22 ± 2 °C) under consistent environmental conditions. A series of filaments with increasing bending forces (approximately 4, 6, 8, 10, 15, and 26 g) was applied perpendicularly to the plantar surface of each hind paw. Each filament was applied three times, with sufficient intervals between stimulations to avoid sensitization. A brisk paw withdrawal of the injured (CCI) limb was considered a positive response. The paw-withdrawal threshold (PWT) was expressed in grams. To minimize stress-related variability, animals were habituated to the testing room and apparatus for two consecutive days prior to baseline measurements. In addition to the ≥60 min acclimation period on test days, animals were placed daily in the testing chambers during the habituation period. All behavioral sessions were video-recorded using a high-definition camera and analyzed offline. Two independent experimenters, blinded to group allocation, reviewed the recordings. Paw-withdrawal thresholds were determined based on consensus scoring between the blinded evaluators.

#### 4.5.3. Sciatic Functional Index (SFI)

Sciatic nerve function was quantified using the SFI (Bain et al., 1989) [31]. Rats were held by the chest, and their hind feet were pressed onto a stamp pad soaked with water-soluble blue ink. They were immediately allowed to walk along a confined walkway 7.5 cm wide and 60 cm long with a dark shelter at the end of the corridor, with the rats leaving their footprints on a piece of paper cut to the appropriate dimensions and placed on the floor of the corridor. From each print, three distances were measured: print length (PL), toe spread (TS; digits 1–5), and intermediary toe spread (IT; digits 2–4), for both the experimental (E) and normal (N) paws. Factors were computed as PLF = (EPL−NPL)/NPL, TSF = (ETS−NTS)/NTS, ITF = (EIT−NIT)/NIT, and SFI was calculated:SFI = −38.3·PLF + 109.5·TSF + 13.3·ITF − 8.8.

Values near 0 indicate normal function; −100 denotes severe impairment.

### 4.6. Electrophysiology

After day-21 behavioral testing, animals were anesthetized (chloral hydrate, 300 mg/kg, i.p.) for nerve conduction studies. Body temperature was maintained at ~37 °C throughout.

#### 4.6.1. Motor Nerve Conduction Velocity (MNCV)

MNCV was assessed using a student physiograph (INCO Pvt. Ltd., Ambala, India) following methods adapted from Saini et al. (2007) [32], with additional precautions to ensure reliability. Recordings were performed in a dedicated electrophysiology facility within a noise-shielded Faraday cage, under stable chloral hydrate anesthesia (350 mg/kg, i.p.), with depth of anesthesia monitored continuously by respiratory rate and reflex testing. Body temperature was maintained at 37 ± 0.5 °C using a heating pad with rectal probe monitoring. Bipolar needle electrodes (26½-gauge) were positioned for supramaximal stimulation (3 V, 0.2 ms, square pulses) at the sciatic notch (proximal site) and knee (distal site). Compound muscle action potentials were recorded from intrinsic foot muscles with surface electrodes, and a ground electrode was placed on the calf. Distances between proximal and distal stimulation sites were measured precisely using digital calipers. Latency values (ms) were determined from stimulus onset to response onset. For each animal, three consecutive trials were recorded and averaged. MNCV (m/s) was calculated as the distance between stimulation sites divided by the latency difference. All analyses were performed offline by an investigator blinded to treatment allocation.

#### 4.6.2. Sensory Nerve Conduction Velocity (SNCV)

SNCV was measured orthodromically following Kurokawa et al. (2004) [33], under the same anesthetic and environmental conditions described above. Thin ring stimulating electrodes were placed on the third digital nerve, and supramaximal square pulses (0.05 ms duration) were delivered at 1 Hz in trains of 10 pulses. Recording electrodes were inserted subcutaneously at the ankle (behind the medial malleolus) with a reference electrode 1 cm proximal. The inter-electrode distance was measured precisely with calipers (approximately 25 mm). Sensory nerve action potentials were recorded, and onset latency was measured. SNCV (m/s) was calculated as the conduction distance divided by onset latency. At least three trials per animal were recorded, averaged, and analyzed offline by a blinded investigator to minimize observer bias.

### 4.7. Biochemical Assays

Animals were sacrificed approximately 1 h after electrophysiological assessment on day 21 by CO_2_ inhalation in accordance with IACUC guidelines. The left sciatic nerve and surrounding perineural tissue were rapidly excised, rinsed with ice-cold isotonic saline, and processed immediately. Nerve tissues were homogenized (10% *w*/*v*) in ice-cold 0.1 M Tris–HCl buffer (pH 7.4) containing appropriate protease inhibitors, kept on ice for 30 min, and centrifuged at 2000× *g* for 10 min at 4 °C. Supernatants were collected for biochemical analyses.

Perineural tissue samples were homogenized separately and centrifuged at 5000× *g* for 10 min at 4 °C; the resulting pellets were retained for myeloperoxidase (MPO) activity assays. Unless otherwise specified, all chemicals and reagents were of analytical grade and purchased from standard commercial suppliers. Absorbance measurements were performed using a microplate reader or spectrophotometer calibrated according to the manufacturer’s instructions. All biochemical parameters were normalized to total protein content and expressed relative to protein concentration.

#### 4.7.1. Total Protein

Total protein concentration was determined using the Lowry method [34], with bovine serum albumin (BSA) as the standard. Absorbance was measured at 750 nm.

#### 4.7.2. Oxidative and Nitrosative Stress Markers

Nitrite (NO metabolites). Nitrite levels were quantified as an index of nitric oxide production following deproteinization with sodium hydroxide and zinc sulfate. After centrifugation, supernatants were incubated with vanadium (III) chloride at 37 °C for 45 min, and absorbance was measured at 540 nm. Sodium nitrite was used to generate the standard curve, and results are expressed as nmol/mg protein [35].

Malondialdehyde (MDA). Lipid peroxidation was assessed by measuring thiobarbituric acid–reactive substances (TBARS) as described previously [36]. Absorbance was recorded at 532 nm, and MDA concentrations were calculated using tetramethoxypropane as the standard. Data are expressed as nmol/mg protein.

Glutathione (GSH). Reduced glutathione levels were determined using Ellman’s reagent (DTNB) following protein precipitation with trichloroacetic acid [37]. Absorbance was measured at 412 nm, and results are expressed as µg/mg protein.

Superoxide Dismutase (SOD). SOD activity was measured based on inhibition of nitro blue tetrazolium reduction in the xanthine/xanthine oxidase system [38]. One unit of SOD activity was defined as the amount of enzyme causing 50% inhibition of the reaction. Results are expressed as U/mg protein.

Catalase (CAT). CAT activity was determined by monitoring the decomposition rate of hydrogen peroxide at 240 nm, as previously described (Doğruer et al., 2004) [39]. One unit of CAT activity was defined as the amount of enzyme decomposing 1 µmol of H_2_O_2_ per minute at 25 °C. Results are expressed as U/mg protein [40].

#### 4.7.3. Pro-Inflammatory Cytokines

Tumor necrosis factor-α (TNF-α), interleukin-1β (IL-1β), and interleukin-6 (IL-6) concentrations were quantified using commercially available sandwich ELISA kits according to the manufacturers’ instructions (e.g., TNF-α, Invitrogen, Waltham, MA, USA; IL-1β and IL-6, Thermo Scientific, Waltham, MA, USA). Cytokine levels were normalized to total protein content and expressed as pg/mg protein.

#### 4.7.4. Myeloperoxidase (MPO)

MPO activity, an index of neutrophil infiltration, was measured following the method described by Bradley et al. [41]. Tissue pellets were resuspended in phosphate buffer containing hexadecyltrimethylammonium bromide, subjected to freeze–thaw cycles with sonication, and centrifuged. Supernatants were incubated with o-dianisidine and hydrogen peroxide, and changes in absorbance at 460 nm were recorded. Results are expressed as mU/mg protein.

#### 4.7.5. Apoptosis Marker (Caspase-3)

Caspase-3 activity was assessed colorimetrically using Ac-DEVD-pNA as a substrate, following the manufacturer’s protocol (R&D Systems/BioPacific, Emeryville, CA, USA). Absorbance was measured at 405 nm, and activity reflects executioner caspase activation associated with apoptotic signaling.

### 4.8. Statistical Analysis

Data are presented as mean ± SEM. Statistical analyses were performed using GraphPad Prism version 8.3.0 (GraphPad Software, San Diego, CA, USA). Prior to parametric analyses, data distributions were assessed for normality using the Shapiro–Wilk test, and no significant deviations from normality were detected.

Behavioral outcomes, including paw-withdrawal latency, paw-withdrawal threshold, and sciatic functional index, were analyzed using repeated-measures two-way ANOVA, with treatment group as the between-subjects factor and time (Day −1, 3, 7, 14, and 21) as the within-subjects factor. When significant main effects or interactions were observed, Tukey’s post hoc test was applied for multiple comparisons. When appropriate, corrections for violations of sphericity were applied. Biochemical and electrophysiological parameters measured at a single time point (including MNCV, SNCV, nitrite, MDA, GSH, SOD, CAT, cytokines, and caspase-3 activity) were analyzed using one-way ANOVA followed by Tukey’s post hoc test. Each experimental group consisted of n = 8 animals, unless otherwise stated. Statistical significance was defined as *p* < 0.05. Detailed statistical outputs are provided in the figure legends.

## 5. Conclusions

In summary, this study demonstrates that geraniin exerts significant neuroprotective and antinociceptive effects in a rat model of CCI-induced neuropathic pain. GRN alleviated pain-related behaviors and improved peripheral nerve function, concomitant with modulation of NO-related signaling, restoration of redox balance, and suppression of neuroinflammatory and apoptotic processes. These findings identify GRN as a promising multitarget natural compound for neuropathic pain and provide a foundation for further pharmacokinetic, mechanistic, and translational studies.

## Figures and Tables

**Figure 1 ijms-27-00507-f001:**
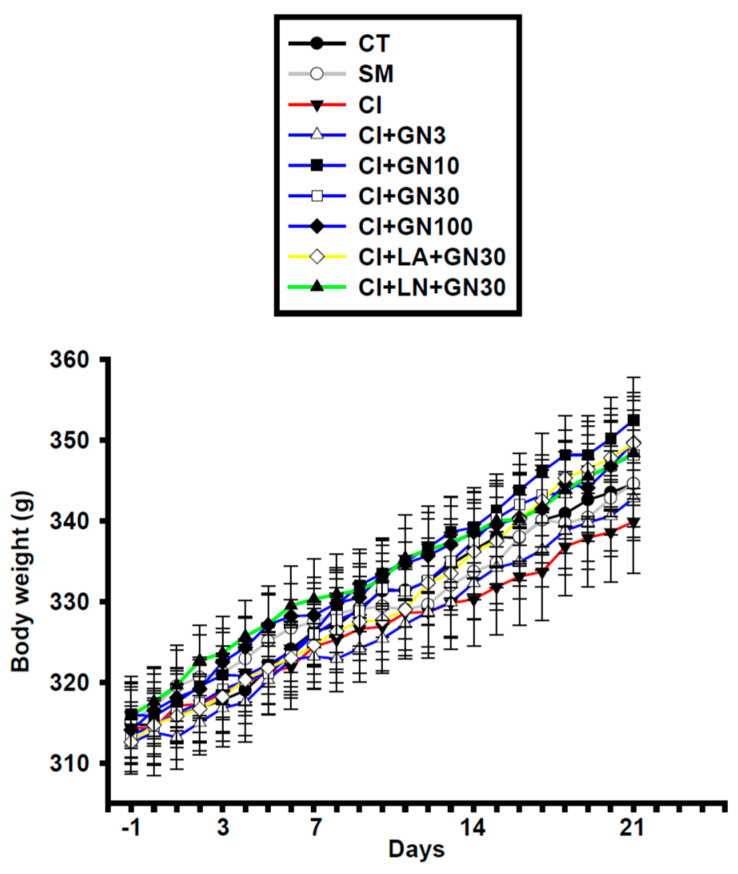
Daily body weight changes in rats throughout the 21-day treatment period. Values are mean ± SEM (n = 8). Data were analyzed by repeated-measures two-way ANOVA (Group × Day) followed by Tukey’s HSD post hoc test. (CT: control group; SM: sham group; CI: CCI group; CI+GN3: CCI with GRN 3 mg/kg treatment group; CI+GN10: CCI with GRN 10 mg/kg treatment group; CI+GN30: CCI with GRN 30 mg/kg treatment group; CI+GN100: CCI with GRN 100 mg/kg treatment group; CI+LA+GN30: CCI with LA 100 mg/kg plus GRN 30 mg/kg treatment group; CI+LN+GN30: CCI with LN 10 mg/kg plus GRN 30 mg/kg treatment group; g: grams).

**Figure 2 ijms-27-00507-f002:**
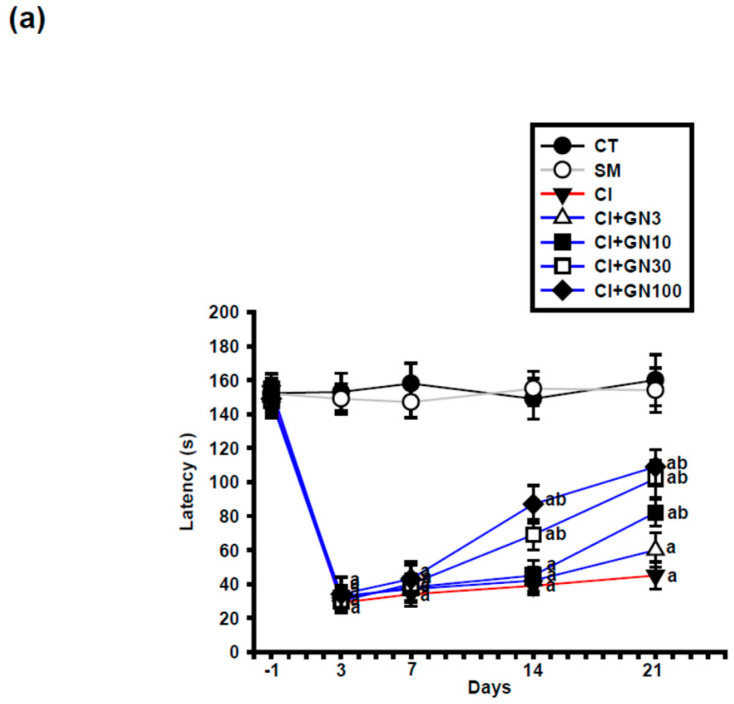
GRN (3–100 mg/kg) alleviates CCI-evoked heat hypersensitivity. Values are mean ± SEM (n = 8). (**a**) Time course of paw-withdrawal latency from baseline (day −1) to day 21. Data were analyzed by repeated-measures two-way ANOVA (Group × Day) followed by Tukey’s HSD post hoc test; (**b**) Improvement (%) at day 21 for each GRN dose. Statistical comparisons were performed using one-way ANOVA followed by Tukey’s post hoc test. ^a^ *p* < 0.001 vs. CT; ^b^ *p* < 0.001 vs. CI. (CT: control group; SM: sham group; CI: CCI group; CI+GN3: CCI with GRN 3 mg/kg treatment group; CI+GN10: CCI with GRN 10 mg/kg treatment group; CI+GN30: CCI with GRN 30 mg/kg treatment group; CI+GN100: CCI with GRN 100 mg/kg treatment group; %: percentage; s: seconds).

**Figure 3 ijms-27-00507-f003:**
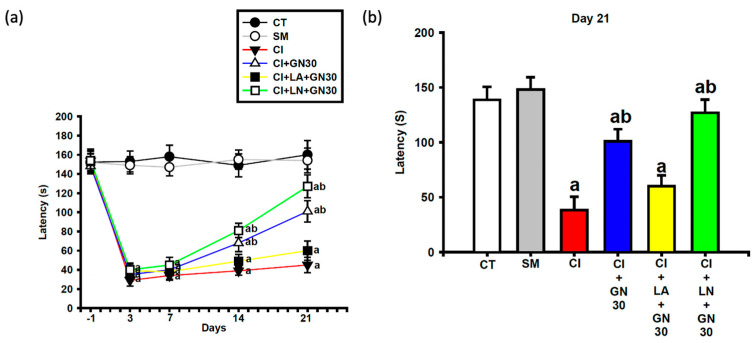
Interaction of GRN (30 mg/kg) with nitric-oxide modulators on thermal hypersensitivity after CCI. Values are expressed as mean ± SEM (n = 8). (**a**) Time course of paw-withdrawal latency from baseline (Day −1) to Day 21. (**b**) Latency at Day 21. Data were analyzed by repeated-measures two-way ANOVA (Group × Day) followed by Tukey’s HSD post hoc test. ^a^
*p* < 0.001 vs. CT; ^b^
*p* < 0.001 vs. CI. (CT: control group; SM: sham group; CI: CCI group; CI+GN30: CCI with GRN 30 mg/kg treatment group; CI+LA+GN30: CCI with LA 100 mg/kg plus GRN 30 mg/kg treatment group; CI+LN+GN30: CCI with LN 10 mg/kg plus GRN 30 mg/kg treatment group; s: seconds).

**Figure 4 ijms-27-00507-f004:**
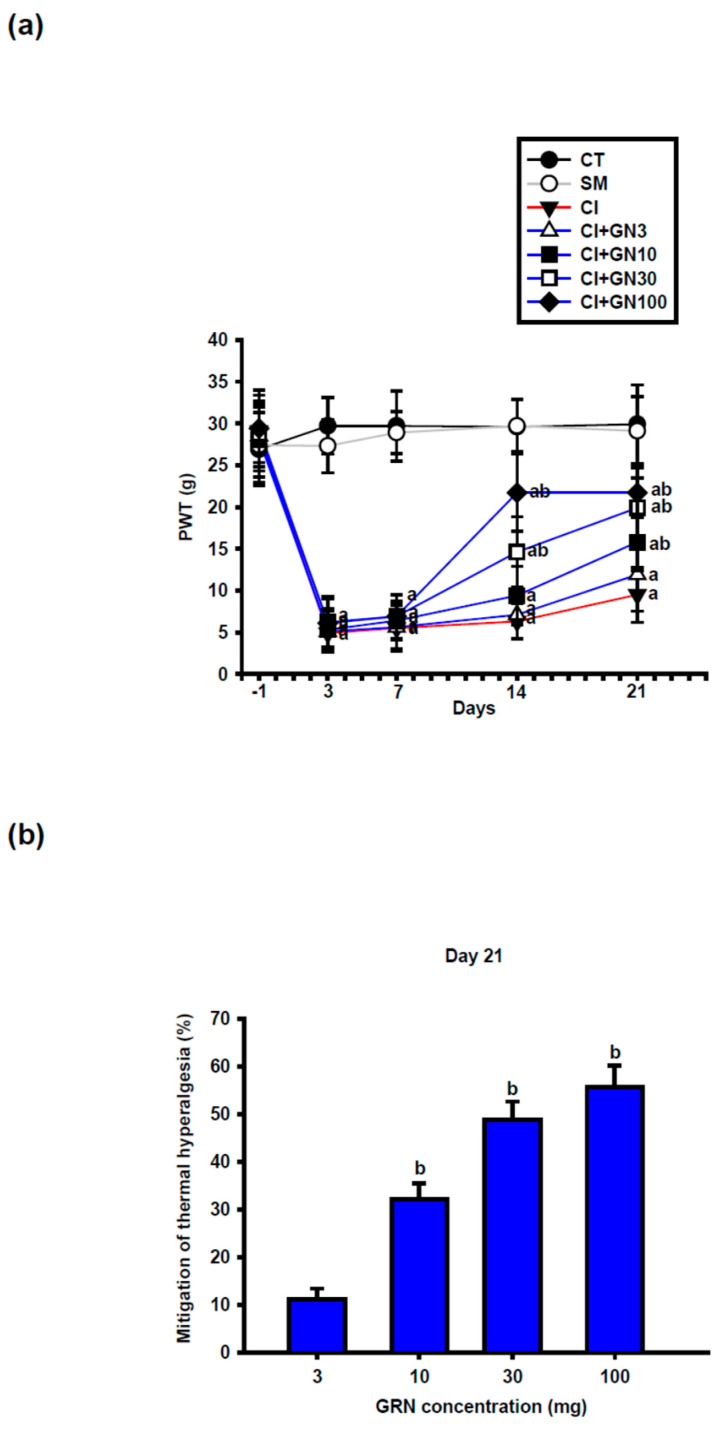
GRN (3–100 mg/kg) counteracts CCI-induced mechanical allodynia. Values are expressed as mean ± SEM (n = 8). (**a**) Time course of paw-withdrawal threshold (PWT) from baseline (Day −1) to Day 21. Data were analyzed by repeated-measures two-way ANOVA (Group × Day) followed by Tukey’s HSD post hoc test. (**b**) Recovery (%) in PWT at Day 21 for each GRN dose. Statistical comparisons were performed using one-way ANOVA followed by Tukey’s post hoc test. ^a^
*p* < 0.001 vs. CT; ^b^
*p* < 0.001 vs. CI. (CT: control group; SM: sham group; CI: CCI group; CI+GN3: CCI with GRN 3 mg/kg treatment group; CI+GN10: CCI with GRN 10 mg/kg treatment group; CI+GN30: CCI with GRN 30 mg/kg treatment group; CI+GN100: CCI with GRN 100 mg/kg treatment group; %: percentage; g: grams).

**Figure 5 ijms-27-00507-f005:**
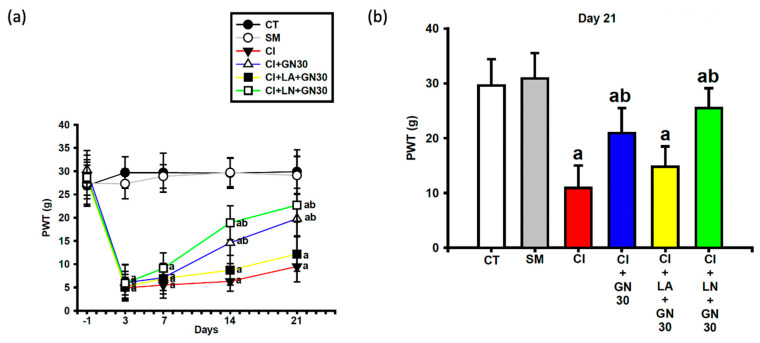
Effects of GRN (30 mg/kg) and NO pathway modulation on mechanical allodynia following CCI. Values are expressed as mean ± SEM (n = 8). (**a**) Time course of paw-withdrawal threshold (PWT) from baseline (Day −1) to Day 21. (**b**) PWT at Day 21. Data were analyzed by repeated-measures two-way ANOVA (Group × Day) followed by Tukey’s HSD post hoc test. ^a^
*p* < 0.001 vs. CT; ^b^
*p* < 0.001 vs. CI. (CT: control group; SM: sham group; CI: CCI group; CI+GN30: CCI with GRN 30 mg/kg treatment group; CI+LA+GN30: CCI with LA 100 mg/kg plus GRN 30 mg/kg treatment group; CI+LN+GN30: CCI with LN 10 mg/kg plus GRN 30 mg/kg treatment group; g: grams).

**Figure 6 ijms-27-00507-f006:**
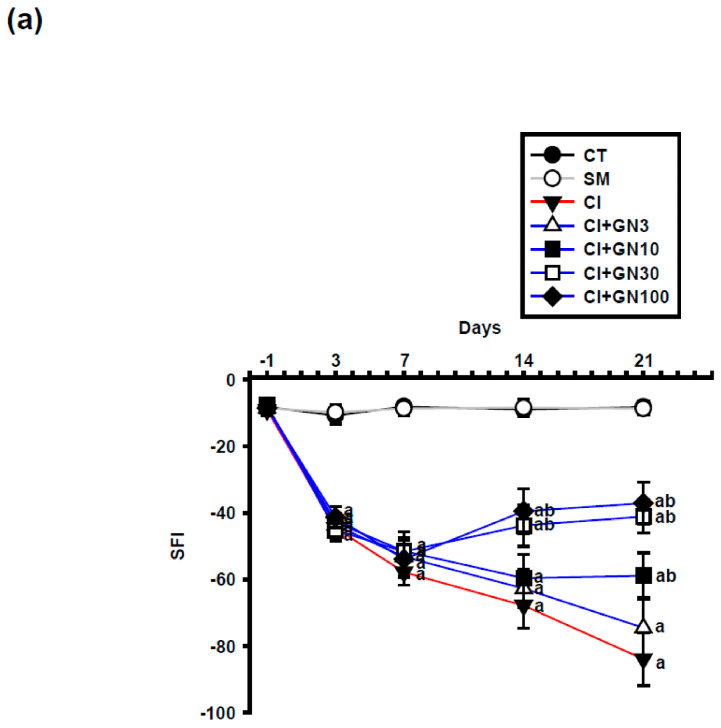
GRN (3–100 mg/kg) improves sciatic nerve function after CCI. Values are expressed as mean ± SEM (n = 8). (**a**) Time course of Sciatic Functional Index (SFI) from baseline (Day −1) through Day 21. Data were analyzed by repeated-measures two-way ANOVA (Group × Day) followed by Tukey’s HSD post hoc test. (**b**) SFI improvement (%) at Day 21 for each GRN dose. Statistical comparisons were performed using one-way ANOVA followed by Tukey’s post hoc test. ^a^
*p* < 0.001 vs. CT; ^b^
*p* < 0.001 vs. CI. (CT: control group; SM: sham group; CI: CCI group; CI+GN3: CCI with GRN 3 mg/kg treatment group; CI+GN10: CCI with GRN 10 mg/kg treatment group; CI+GN30: CCI with GRN 30 mg/kg treatment group; CI+GN100: CCI with GRN 100 mg/kg treatment group; %: percentage).

**Figure 7 ijms-27-00507-f007:**
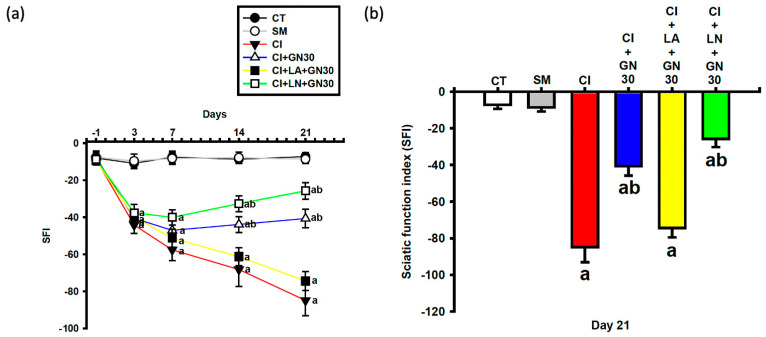
GRN (30 mg/kg) and NO modulators influence sciatic functional outcomes after CCI. Values are expressed as mean ± SEM (n = 8). (**a**) Time course of Sciatic Functional Index (SFI) from baseline (Day −1) to Day 21. (**b**) SFI values at Day 21. Data were analyzed by repeated-measures two-way ANOVA (Group × Day) followed by Tukey’s HSD post hoc test. ^a^
*p* < 0.001 vs. CT; ^b^
*p* < 0.001 vs. CI. (CT: control group; SM: sham group; CI: CCI group; CI+GN30: CCI with GRN 30 mg/kg treatment group; CI+LA+GN30: CCI with LA 100 mg/kg plus GRN 30 mg/kg treatment group; CI+LN+GN30: CCI with LN 10 mg/kg plus GRN 30 mg/kg treatment group).

**Figure 8 ijms-27-00507-f008:**
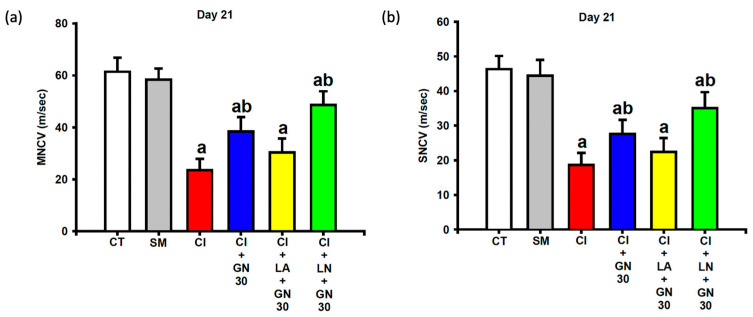
GRN (30 mg/kg) and NO modulators ameliorate CCI-related nerve conduction deficits at day 21. Values are expressed as mean ± SEM (n = 8). (**a**) Motor nerve conduction velocity (MNCV). (**b**) Sensory nerve conduction velocity (SNCV). Data were analyzed by one-way ANOVA followed by Tukey’s HSD post hoc test. ^a^
*p* < 0.001 vs. CT; ^b^
*p* < 0.001 vs. CI. (CT: control group; SM: sham group; CI: CCI group; CI+GN30: CCI with GRN 30 mg/kg treatment group; CI+LA+GN30: CCI with LA 100 mg/kg plus GRN 30 mg/kg treatment group; CI+LN+GN30: CCI with LN 10 mg/kg plus GRN 30 mg/kg treatment group).

**Figure 9 ijms-27-00507-f009:**
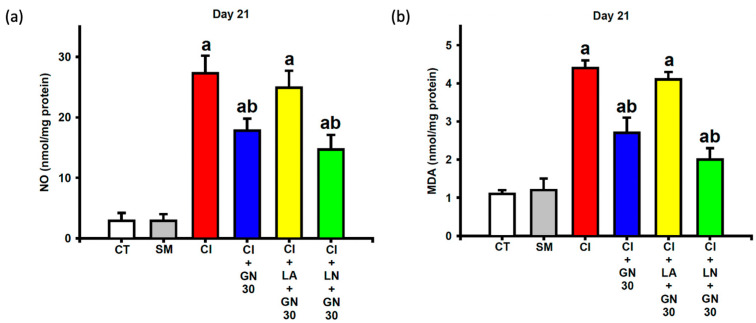
GRN (30 mg/kg) and nitric-oxide modulators reduce nitrosative and oxidative stress in sciatic nerves at Day 21 after CCI. Values are expressed as mean ± SEM (n = 8). (**a**) Nitric-oxide metabolites (NO). (**b**) Malondialdehyde (MDA). Data were analyzed by one-way ANOVA followed by Tukey’s HSD post hoc test. ^a^
*p* < 0.001 vs. CT; ^b^
*p* < 0.001 vs. CI. (CT: control group; SM: sham group; CI: CCI group; CI+GN30: CCI with GRN 30 mg/kg treatment group; CI+LA+GN30: CCI with LA 100 mg/kg plus GRN 30 mg/kg treatment group; CI+LN+GN30: CCI with LN 10 mg/kg plus GRN 30 mg/kg treatment group).

**Figure 10 ijms-27-00507-f010:**
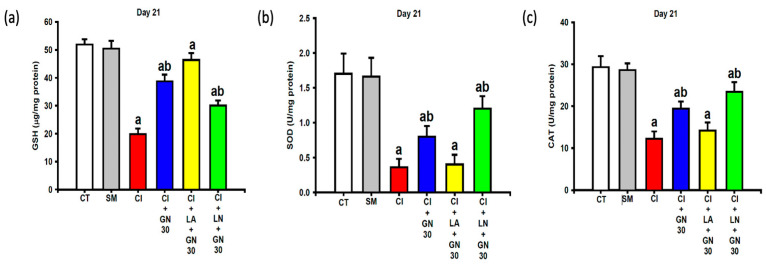
Restoration of antioxidant defenses by GRN (30 mg/kg) and nitric-oxide modulators at Day 21 post-CCI. Values are expressed as mean ± SEM (n = 8). (**a**) Glutathione (GSH). (**b**) Superoxide dismutase (SOD). (**c**) Catalase (CAT). Data were analyzed by one-way ANOVA followed by Tukey’s HSD post hoc test. ^a^
*p* < 0.001 vs. CT; ^b^
*p* < 0.001 vs. CI. (CT: control group; SM: sham group; CI: CCI group; CI+GN30: CCI with GRN 30 mg/kg treatment group; CI+LA+GN30: CCI with LA 100 mg/kg plus GRN 30 mg/kg treatment group; CI+LN+GN30: CCI with LN 10 mg/kg plus GRN 30 mg/kg treatment group).

**Figure 11 ijms-27-00507-f011:**
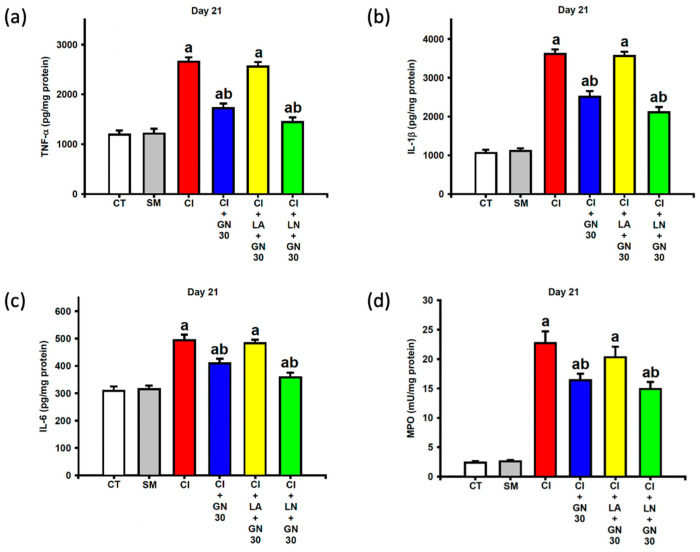
GRN (30 mg/kg) and nitric-oxide modulators attenuate neuroinflammatory mediators in sciatic nerve tissue at Day 21 post-CCI. Values are expressed as mean ± SEM (n = 8). (**a**) Tumor necrosis factor-α (TNF-α). (**b**) Interleukin-1β (IL-1β). (**c**) Interleukin-6 (IL-6). (**d**) Myeloperoxidase (MPO). Data were analyzed by one-way ANOVA followed by Tukey’s HSD post hoc test. ^a^
*p* < 0.001 vs. CT; ^b^
*p* < 0.001 vs. CI. (CT: control group; SM: sham group; CI: CCI group; CI+GN30: CCI with GRN 30 mg/kg treatment group; CI+LA+GN30: CCI with LA 100 mg/kg plus GRN 30 mg/kg treatment group; CI+LN+GN30: CCI with LN 10 mg/kg plus GRN 30 mg/kg treatment group).

**Figure 12 ijms-27-00507-f012:**
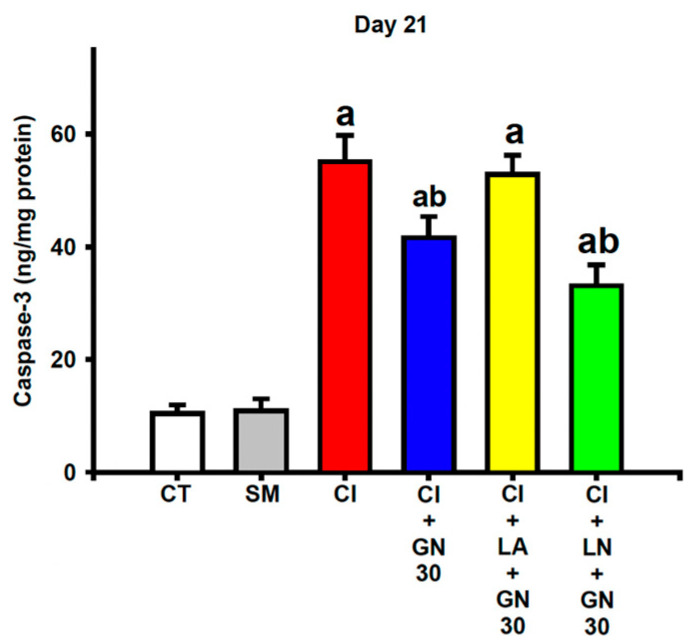
GRN (30 mg/kg) and nitric-oxide modulators downregulate apoptosis-related signaling in sciatic nerves at Day 21 after CCI. Values are expressed as mean ± SEM (n = 8). Caspase-3 levels in sciatic nerve tissue. Data were analyzed by one-way ANOVA followed by Tukey’s HSD post hoc test. ^a^ *p* < 0.001 vs. CT; ^b^ *p* < 0.001 vs. CI. (CT: control group; SM: sham group; CI: CCI group; CI+GN30: CCI with GRN 30 mg/kg treatment group; CI+LA+GN30: CCI with LA 100 mg/kg plus GRN 30 mg/kg treatment group; CI+LN+GN30: CCI with LN 10 mg/kg plus GRN 30 mg/kg treatment group).

**Figure 13 ijms-27-00507-f013:**
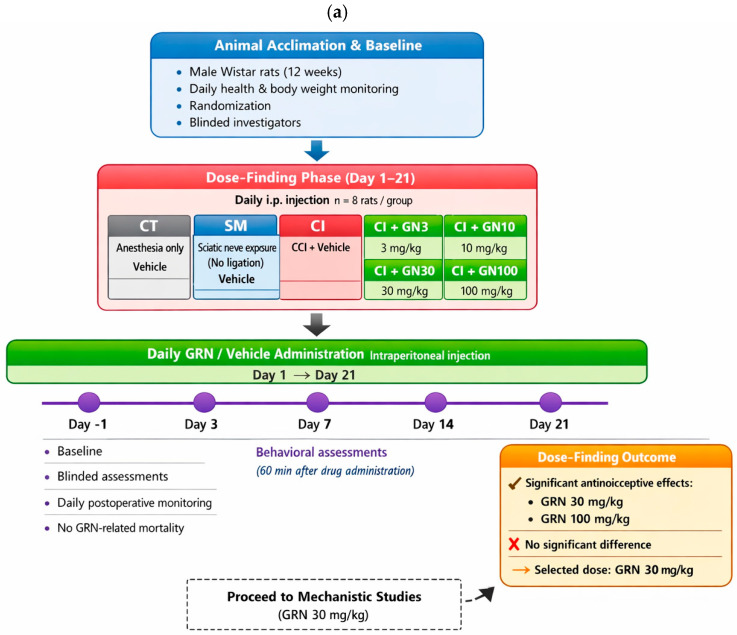
Schematic overview of the experimental design, illustrating group allocation, drug administration schedules, behavioral testing timeline, and tissue collection points. (**a**) Dose-finding phase for behavioral assessment. (**b**) Mechanistic follow-up phase incorporating behavioral, electrophysiological, and biochemical endpoints. CCI, chronic constriction injury; Sh, sham; i.p., intraperitoneal.

## Data Availability

The datasets generated and analyzed during the current study are available from the corresponding author upon reasonable request. Please note that these data are not publicly accessible in order to maintain confidentiality and protect participant privacy.

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
