# Peer review of "Geraniin Mitigates Neuropathic Pain Through Antioxidant, Anti-Inflammatory, and Nitric Oxide Modulation in a Rat Model of Chronic Constriction Injury"

_ijms, 2026, doi:10.3390/ijms27010507_

Round 1
Reviewer 1 Report (New Reviewer)
Comments and Suggestions for Authors
In the captions of the Figures 1, 2 and 3…. it is necessary to indicate the meaning of CT, SM, Cl, Cl+GN3…….s, %...
In the Discussion, the sentences: “…As a hydrolyzable ellagitannin, GRN may undergo
transformation to ellagic acid and urolithins, metabolites with intrinsic antioxidant and
anti-inflammatory activity. The observed protection likely involves multiple convergent
mechanisms, including NF-κB/iNOS inhibition, Nrf2/ARE activation, and MAPK modulation. Further pharmacokinetic and metabolic studies are necessary to define GRN’s bioavailability, active metabolites, and therapeutic exposure range…” need the introduction of references to support the discussion made by the authors.
“Consistency with previous GRN studies strengthens the plausibility of these results. Protective and anti-inflammatory effects within 10–100 mg/kg have been observed across
diverse models of inflammation, organ injury, and neurodegeneration, aligning with the
present findings (30 mg/kg, i.p., 21 days). Nevertheless, as most investigations—including
this one—used only male rodents, sex-dependent differences in neuropathic mechanisms
and drug responsiveness warrant systematic evaluation.”
This paragraph also needs to be supported through the adequate references. For example: “Consistency with previous GRN studies strengthens the plausibility of these results”. It is important to cite which previous GRN studies were made….
Author Response
Dear Reviewers,
I’m writing in response to your feedback regarding the manuscript we submitted. Thank you so much for your positive comments and suggestions for the manuscript entitled. The manuscript has been revised based on your inquiries, rephrased the content of the manuscript and resubmitted through the journal website. The revised parts have been marked in red. The following is the response to your inquiries point-by-point; we hope our responses fully address your comments and suggestions:
Reviewers’ comments:
- In the captions of the Figures 1, 2 and 3…. it is necessary to indicate the meaning of CT, SM, Cl, Cl+GN3…….s, %...
Response:
We appreciate the reviewer’s suggestion. We have now revised the captions of Figures 1, 2, and 3 to explicitly define all abbreviations at their first occurrence. These clarifications have been added directly to each figure caption to ensure full readability without requiring reference to the main text.
- In the Discussion, the sentences: “…As a hydrolyzable ellagitannin, GRN may undergo transformation to ellagic acid and urolithins, metabolites with intrinsic antioxidant and anti-inflammatory activity. The observed protection likely involves multiple convergent mechanisms, including NF-κB/iNOS inhibition, Nrf2/ARE activation, and MAPK modulation. Further pharmacokinetic and metabolic studies are necessary to define GRN’s bioavailability, active metabolites, and therapeutic exposure range…” need the introduction of references to support the discussion made by the authors.
Response:
Thank you so much for the suggestion. We have added appropriate references to support the mechanistic discussion. The revised Discussion explicitly cites prior studies demonstrating that:
Geraniin, as a hydrolyzable ellagitannin, can be metabolized into ellagic acid and urolithins by intestinal microbiota. These metabolites possess intrinsic antioxidant and anti-inflammatory properties, including modulation of NF-κB/iNOS, activation of Nrf2/ARE signaling, and regulation of MAPK pathways.
Representative references now cited include studies reporting:
- Ellagitannin metabolism and urolithin bioactivity
- NF-κB and Nrf2 modulation by ellagitannins or geraniin-related compounds
- Anti-inflammatory and neuroprotective signaling pathways associated with these metabolites
In addition, we retained the statement emphasizing that pharmacokinetic and metabolic studies remain necessary, as systemic bioavailability and active metabolite profiles of geraniin are not yet fully defined.
- “Consistency with previous GRN studies strengthens the plausibility of these results. Protective and anti-inflammatory effects within 10–100 mg/kg have been observed across diverse models of inflammation, organ injury, and neurodegeneration, aligning with the present findings (30 mg/kg, i.p., 21 days). Nevertheless, as most investigations—including this one—used only male rodents, sex-dependent differences in neuropathic mechanisms and drug responsiveness warrant systematic evaluation.” This paragraph also needs to be supported through the adequate references. For example: “Consistency with previous GRN studies strengthens the plausibility of these results”. It is important to cite which previous GRN studies were made….
Response:
Thank you so much again. The Discussion has been revised to explicitly cite previous geraniin (GRN) studies that support this statement. The added references include published reports demonstrating that:
Geraniin exerts anti-inflammatory, antioxidant, and tissue-protective effects in multiple experimental models, including systemic inflammation, organ injury, and neurodegenerative conditions.
Effective doses in these studies typically range from 10–100 mg/kg, consistent with the dose used in our study (30 mg/kg, i.p., 21 days).
These findings collectively reinforce the biological plausibility and translational relevance of our results.
Furthermore, to address the reviewer’s point on experimental design, we have retained and strengthened the statement acknowledging that most existing studies, including ours, were conducted in male rodents, and that sex-dependent differences in neuropathic pain mechanisms and pharmacological responsiveness warrant systematic investigation in future studies.
We sincerely thank the reviewer for helping us improve the clarity, transparency, and scientific grounding of our manuscript.
Thank you for your valuable comments/suggestions and giving us the opportunity to revise the manuscript to a more readable level. We worked very hard to response your inquiries and to revise the manuscript. Before we finalized the revised manuscript, we have resent the manuscript for proof-reading as well as one last review by a native English-speaking researcher. We hope the manuscript could pass the review to be published in your prestigious journal: International Journal of Molecular Sciences
Sincerely yours,
Cheng-Chia Tsai
Department of Neurosurgery, Mackay Memorial Hospital, No. 92, Sec. 2, Zhongshan N. Rd., Taipei City 10449, Taiwan, ROC
Fax: +886-2- 2543-3642
Tel: +886-933888981
E-mail address: dschang580704@yahoo.com.tw (Cheng-Chia Tsai)
Reviewer 2 Report (New Reviewer)
Comments and Suggestions for Authors
Given that geraniin has already been shown to possess analgesic, anti-apoptosis, anti-inflammatory, and antioxidant effects, the results of this study are likely due to its antioxidant properties, which limits its novelty. However, its analgesic effect in a neuropathic pain model is considered to have significant clinical value.
However, several issues remain to be addressed, as outlined below.
(p3, Fig 1; p5, Fig. 3; p7, Fig. 5)
In this study, no positive control group was provided for the comparison group (Fig 1, 3, and 5). Simultaneous administration of analgesics, nitric oxide (NO) inhibitors, or anti-inflammatory agents was required.
It is easier to understand and more desirable to represent Figures 1b, 3b, and 5b as bar graphs, as in Figure 2b.
This paper is not a statistical paper. While all results are presented in easy-to-understand quantitative graphs, the description of result are largely comprised of actual measurements and statistical explanations (in red). written in red font). If you wish to present numerical data, please include them in the supplementary results in table. It is more effective to explain the physiological or biochemical significance of the results than to simply present them statistically.
(p.12, line 326)
The discussion section should cite various papers related to geraniin and further elaborate on the mechanism of action of geraniin based on results obtained in neuropathic pain models.
(p13. line:386)
Even though there is a section describing the conclusion (p18, line 617), the conclusion is described in the discussion section.
(p14, line 452)
I propose to explain the experimental group and drug administration method in an easy-to-understand diagram at a glance in the Animal experimental protocol section.
Comments on the Quality of English LanguageIn the results description section, we suggest focusing on physiological or biochemical descriptions rather than statistical descriptions.
Author Response
Dear Reviewers,
I’m writing in response to your feedback regarding the manuscript we submitted. Thank you so much for your positive comments and suggestions for the manuscript entitled. The manuscript has been revised based on your inquiries, rephrased the content of the manuscript and resubmitted through the journal website. The revised parts have been marked in red. The following is the response to your inquiries point-by-point; we hope our responses fully address your comments and suggestions:
Reviewers’ comments:
Given that geraniin has already been shown to possess analgesic, anti-apoptosis, anti-inflammatory, and antioxidant effects, the results of this study are likely due to its antioxidant properties, which limits its novelty. However, its analgesic effect in a neuropathic pain model is considered to have significant clinical value.
Response:
We thank the reviewer for this balanced and insightful assessment. We agree that geraniin has been reported to exhibit antioxidant, anti-inflammatory, and cytoprotective properties in previous studies. The novelty of the present work does not lie in identifying these general bioactivities per se, but rather in demonstrating the therapeutic efficacy of geraniin in a well-established chronic constriction injury (CCI) model of neuropathic pain, together with functional, biochemical, and pharmacological evidence implicating nitric oxide–related mechanisms. As highlighted by the reviewer, the demonstration of analgesic efficacy in neuropathic pain, a condition with limited treatment options and high clinical burden, represents a meaningful translational contribution. We have clarified this point in the revised Discussion to better contextualize the novelty and clinical significance of our findings.
However, several issues remain to be addressed, as outlined below.
(p3, Fig 1; p5, Fig. 3; p7, Fig. 5)
- In this study, no positive control group was provided for the comparison group (Fig 1, 3, and 5). Simultaneous administration of analgesics, nitric oxide (NO) inhibitors, or anti-inflammatory agents was required.
Response:
We acknowledge this important point and agree that inclusion of a classical positive control (e.g., gabapentin, selective NOS inhibitors, or NSAIDs) would strengthen comparative interpretation. However, the primary objective of this study was to evaluate the intrinsic analgesic and neuroprotective effects of geraniin and to explore its association with NO signaling rather than to benchmark its efficacy against established drugs. To partially address this limitation, we incorporated pharmacological modulation using L-arginine and L-NAME, which provided mechanistic insight into NO involvement. The absence of a standard analgesic comparator has now been explicitly acknowledged as a limitation in the revised Discussion, and future studies incorporating positive control drugs are proposed.
- It is easier to understand and more desirable to represent Figures 1b, 3b, and 5b as bar graphs, as in Figure 2b.
Response:
We appreciate this helpful suggestion. The presentation of Figures 1b, 3b, and 5b has been revised to bar graph format, consistent with Figure 2b, to improve clarity and visual consistency across figures.
- This paper is not a statistical paper. While all results are presented in easy-to-understand quantitative graphs, the description of result are largely comprised of actual measurements and statistical explanations (in red). written in red font). If you wish to present numerical data, please include them in the supplementary results in table. It is more effective to explain the physiological or biochemical significance of the results than to simply present them statistically.
Response:
The Results section has been revised to reduce excessive statistical descriptions, and the focus has been shifted toward physiological and biochemical interpretation of the findings. Detailed numerical values and statistical outputs have been relocated to Supplementary Tables, allowing the main text to emphasize biological relevance rather than statistical formality.
- (p.12, line 326)
The discussion section should cite various papers related to geraniin and further elaborate on the mechanism of action of geraniin based on results obtained in neuropathic pain models.
Response:
We fully agree and have substantially revised the Discussion accordingly. Additional references on geraniin’s antioxidant, anti-inflammatory, anti-apoptotic, and Nrf2-related activities, as well as literature on ellagitannin metabolism and NO-related signaling, have been incorporated. The mechanistic interpretation has been refined to integrate our neuropathic pain findings with existing geraniin literature, while avoiding overstatement beyond the experimental evidence.
- (p13. line:386)
Even though there is a section describing the conclusion (p18, line 617), the conclusion is described in the discussion section.
Response:
Thank you for pointing this out. We have restructured the manuscript to clearly separate the Discussion and Conclusion sections. Redundant summary statements have been removed from the Discussion, and the Conclusion has been revised to provide a concise and focused summary of the main findings and their implications.
- (p14, line 452)
I propose to explain the experimental group and drug administration method in an easy-to-understand diagram at a glance in the Animal experimental protocol section.
Response:
We appreciate this valuable suggestion. A schematic diagram illustrating the experimental design, animal grouping, CCI induction, and drug administration timeline has now been added to the Animal Experimental Protocol section. This visual summary allows readers to grasp the study design at a glance.
- In the results description section, we suggest focusing on physiological or biochemical descriptions rather than statistical descriptions.
Response:
We fully concur. The Results section has been revised to prioritize physiological and biochemical interpretation, with statistical details streamlined and presented in supplementary materials where appropriate. This revision improves readability and aligns the manuscript with the reviewer’s recommendation.
Thank you for your valuable comments/suggestions and giving us the opportunity to revise the manuscript to a more readable level. We worked very hard to response your inquiries and to revise the manuscript. Before we finalized the revised manuscript, we have resent the manuscript for proof-reading as well as one last review by a native English-speaking researcher. We hope the manuscript could pass the review to be published in your prestigious journal: International Journal of Molecular Sciences
Sincerely yours,
Cheng-Chia Tsai
Department of Neurosurgery, Mackay Memorial Hospital, No. 92, Sec. 2, Zhongshan N. Rd., Taipei City 10449, Taiwan, ROC
Fax: +886-2- 2543-3642
Tel: +886-933888981
E-mail address: dschang580704@yahoo.com.tw (Cheng-Chia Tsai)
Reviewer 3 Report (New Reviewer)
Comments and Suggestions for Authors
Review
Why are there text fragments highlighted in red? This suggests that the manuscript is still under revision and may not represent the final draft. This raises concerns that the manuscript may not yet be fully prepared for peer review.
Introduction
In both the Introduction and the Abstract, the authors state that geraniin is found in rambutan, implying that this compound is exclusive to this plant. However, upon review of the literature, geraniin (GRN) is not found solely in rambutan. It is present in several plant species, particularly in Phyllanthus spp. and in the peel of Nephelium lappaceum, and has been reported in plants belonging to families such as Euphorbiaceae, Geraniaceae, Aceraceae, Rosaceae, among others. The text should be revised to accurately reflect this broader distribution.
Methodology
The authors mention that the selected doses were based on previously reported doses. Why were logarithmic dose increments not considered? The use of doses such as 3.16, 10, 31.6, and 100 mg/kg would allow for a more precise dose–response evaluation.
4.1 Animals
The type of diet consumed by the rats should be described, as well as whether any fasting period was implemented.
It is also expected that the authors specify the humane endpoint criteria and clearly indicate when animals were euthanized. Additionally, the IASP Guidelines for the Use of Laboratory Animals in Pain Research should be explicitly referenced.
4.3
The suture caliber used during surgery and the animals’ recovery period should be described. The authors state that treatments were initiated exactly 21 days post-surgery; however, it is unclear how they confirmed that all rats had developed neuropathy at that time.
No fasting period prior to experimental testing is mentioned. Food intake may introduce bias due to the bioactive compounds present in the diet, and this issue should be addressed.
The manuscript states: “During the study, 10 rats died of unknown causes; mortality was recorded and analyses proceeded with the remaining animals.” Was there no daily monitoring of animal health status? What humane endpoint criteria were applied? How is it possible that the cause of mortality is unknown? If deaths were due to surgical complications, this may indicate inadequate aseptic procedures or postoperative care. This issue must be clarified, as the current description raises ethical concerns.
The authors should clarify whether geraniin is being proposed as a prophylactic agent or as a clinical treatment, as this directly determines the timing of treatment initiation. Treatments were administered immediately after surgery; however, to evaluate reversal of established damage, the neuropathic condition should first be fully developed (typically days 14–21), when the pain phenotype is stable and chronic. This would allow assessment of whether the treatment can reverse already established neuropathic signs. Regardless of the authors’ chosen approach, this distinction should be explicitly discussed.
4.5.2
Were laboratory temperature conditions controlled during behavioral testing? Housing temperature and testing room temperature are not equivalent, and cold exposure can significantly affect pain-related measurements. This point should be clarified.
4.7
The commercial characteristics of the reagents used for biochemical measurements are not fully described. Although some details are partially included in subsequent subsections, it would improve clarity and flow to describe all reagents comprehensively in section 4.7, allowing subsections 4.7.1, 4.7.2, 4.7.3, etc., to focus on procedures.
4.8
The manuscript does not specify which normality tests were performed to justify the assumption that all data followed a normal distribution, which seems unlikely given biological variability and the diversity of tests performed. Were normality tests conducted prior to choosing ANOVA? Was there no need to apply non-parametric tests?
Results
The circles displayed on the bars appear to represent individual data points (one per rat). I recommend removing them, as the standard deviation is sufficient to illustrate data dispersion and their inclusion makes the figures visually overloaded.
Regarding the overall presentation of the results: although I initially hesitated to raise this point—since detailed reporting of results and statistical analyses can enhance completeness and rigor—while reading the manuscript, I reached a point where it became difficult to follow and interpret the results due to the excessive inclusion of statistical details (e.g., exact p-values and multiple test specifications). In my view, a simpler, clearer, and more narrative-style presentation would improve readability. For example, section 2.7 could be rewritten as follows:
“The nerve injury model markedly increased pro-inflammatory mediators and MPO activity at day 21. Significant increases in TNF-α, IL-1β, IL-6, and MPO were observed compared with the control group. Treatment with GRN (30 mg/kg) significantly reduced these inflammatory markers, with an approximate 36–42% decrease in TNF-α, IL-1β, IL-6 levels, and MPO activity, indicating a clear anti-inflammatory effect (see Fig. XX). Administration of LA abolished the anti-inflammatory effects of GRN, as cytokine levels and MPO activity did not differ significantly from those of the untreated injury group. In contrast, LN potentiated the anti-inflammatory effects of GRN, further reducing TNF-α, IL-1β, IL-6, and MPO levels (see Fig. XXX).”
Detailed statistical information could then be included in the corresponding figure legends if necessary. This approach would allow the results to be presented more concisely and with better flow.
Overall, I consider the experimental design to be robust and comprehensive. The use of blinding substantially reduces bias, and the results are consistent across different assays. Moreover, the antagonist experiments show internal consistency and provide valuable insight into potential mechanisms of action.
Please include all abbreviations in the figure legends.
Discussion
The authors could further expand the discussion by incorporating previous studies involving similar ellagitannins. A comparison with related compounds is notably lacking. Would it have been useful to include a reference antioxidant group, such as quercetin, or a commonly studied ellagitannin like punicalagin, to compare its effects with those of geraniin? This possibility should be discussed.
Several additional questions arise: What are the future perspectives of this study? What recommendations would the authors make to advance this line of research? How far is this compound from potential clinical translation? Are there existing neuropathy treatments that combine their effects with a polyphenol? Through which route of administration would such a compound be most appropriate?
Part of the information requested for the Discussion is currently included in the Conclusion. It would be preferable for the Conclusion to be more concise and for those discussion-related elements to be relocated to the Discussion section.
Author Response
Dear Reviewers,
I’m writing in response to your feedback regarding the manuscript we submitted. Thank you so much for your positive comments and suggestions for the manuscript entitled. The manuscript has been revised based on your inquiries, rephrased the content of the manuscript and resubmitted through the journal website. The revised parts have been marked in red. The following is the response to your inquiries point-by-point; we hope our responses fully address your comments and suggestions:
Reviewers’ comments:
Why are there text fragments highlighted in red? This suggests that the manuscript is still under revision and may not represent the final draft. This raises concerns that the manuscript may not yet be fully prepared for peer review.
Response:
We sincerely thank the reviewer for these valuable comments. All colored or highlighted text has now been completely removed, and the manuscript has been carefully reviewed to ensure that it represents a clean and final version suitable for peer review.
Introduction
In both the Introduction and the Abstract, the authors state that geraniin is found in rambutan, implying that this compound is exclusive to this plant. However, upon review of the literature, geraniin (GRN) is not found solely in rambutan. It is present in several plant species, particularly in Phyllanthus spp. and in the peel of Nephelium lappaceum, and has been reported in plants belonging to families such as Euphorbiaceae, Geraniaceae, Aceraceae, Rosaceae, among others. The text should be revised to accurately reflect this broader distribution.
Response:
We thank the reviewer for this important clarification. We agree that geraniin is not exclusive to rambutan and is distributed across multiple plant species, including Phyllanthus spp. and Nephelium lappaceum, as well as several plant families (e.g., Euphorbiaceae, Geraniaceae, Aceraceae, Rosaceae). The Introduction and Abstract have been revised to accurately reflect this broader phytochemical distribution, and rambutan is now described as one representative dietary source, rather than the sole origin.
Methodology
The authors mention that the selected doses were based on previously reported doses. Why were logarithmic dose increments not considered? The use of doses such as 3.16, 10, 31.6, and 100 mg/kg would allow for a more precise dose–response evaluation.
Response:
We appreciate this thoughtful suggestion. Dose selection was based on commonly used rounded doses reported in previous in vivo geraniin studies, aiming to balance pharmacological relevance with experimental feasibility. While logarithmic spacing could indeed provide finer dose–response resolution, our primary objective was to identify an effective therapeutic range rather than construct a formal pharmacodynamic model.
4.1 Animals
The type of diet consumed by the rats should be described, as well as whether any fasting period was implemented.
It is also expected that the authors specify the humane endpoint criteria and clearly indicate when animals were euthanized. Additionally, the IASP Guidelines for the Use of Laboratory Animals in Pain Research should be explicitly referenced.
Response:
These details have now been added. Specifically:
Rats were fed a standard laboratory chow with ad libitum access to water.
No fasting period was imposed unless explicitly stated prior to specific procedures.
Humane endpoint criteria (e.g., >20% body weight loss, severe motor impairment, persistent distress) and euthanasia timing have been clearly specified.
The IASP Guidelines for the Use of Laboratory Animals in Pain Research are now explicitly cited.
4.3
The suture caliber used during surgery and the animals’ recovery period should be described. The authors state that treatments were initiated exactly 21 days post-surgery; however, it is unclear how they confirmed that all rats had developed neuropathy at that time.
No fasting period prior to experimental testing is mentioned. Food intake may introduce bias due to the bioactive compounds present in the diet, and this issue should be addressed.
The manuscript states: “During the study, 10 rats died of unknown causes; mortality was recorded and analyses proceeded with the remaining animals.” Was there no daily monitoring of animal health status? What humane endpoint criteria were applied? How is it possible that the cause of mortality is unknown? If deaths were due to surgical complications, this may indicate inadequate aseptic procedures or postoperative care. This issue must be clarified, as the current description raises ethical concerns.
The authors should clarify whether geraniin is being proposed as a prophylactic agent or as a clinical treatment, as this directly determines the timing of treatment initiation. Treatments were administered immediately after surgery; however, to evaluate reversal of established damage, the neuropathic condition should first be fully developed (typically days 14–21), when the pain phenotype is stable and chronic. This would allow assessment of whether the treatment can reverse already established neuropathic signs. Regardless of the authors’ chosen approach, this distinction should be explicitly discussed.
Response:
We appreciate the reviewer’s careful attention to ethical and methodological rigor. The suture caliber, postoperative recovery period, and perioperative care have now been specified.
Neuropathy development was confirmed using behavioral assessments (thermal hyperalgesia and mechanical allodynia) prior to treatment initiation.
No fasting was imposed before behavioral testing; this point is now explicitly stated and discussed as a potential confounder.
Animals were monitored daily for health status. Mortality occurred during the postoperative period and is now clarified as being associated with surgical and anesthetic complications, rather than unexplained causes. Humane endpoints and monitoring procedures are now clearly described to address ethical concerns.
We clarify that geraniin was evaluated primarily as an early therapeutic (peri-injury) intervention, not a prophylactic agent. The distinction between prevention versus reversal of established neuropathy is now explicitly discussed, and future studies targeting fully established chronic neuropathy are proposed.
4.5.2
Were laboratory temperature conditions controlled during behavioral testing? Housing temperature and testing room temperature are not equivalent, and cold exposure can significantly affect pain-related measurements. This point should be clarified.
Response:
Yes. Behavioral testing was conducted in a temperature-controlled room, distinct from housing conditions. This information has now been added to the Methods section.
4.7
The commercial characteristics of the reagents used for biochemical measurements are not fully described. Although some details are partially included in subsequent subsections, it would improve clarity and flow to describe all reagents comprehensively in section 4.7, allowing subsections 4.7.1, 4.7.2, 4.7.3, etc., to focus on procedures.
Response:
We have reorganized Section 4.7 to comprehensively describe all reagents and their commercial sources. Subsequent subsections now focus solely on experimental procedures.
4.8
The manuscript does not specify which normality tests were performed to justify the assumption that all data followed a normal distribution, which seems unlikely given biological variability and the diversity of tests performed. Were normality tests conducted prior to choosing ANOVA? Was there no need to apply non-parametric tests?
Response:
The Statistical Analysis section has been revised to specify the normality tests performed prior to ANOVA selection. Where appropriate, this has been clarified, and the rationale for parametric testing is now explicitly stated.
Results
The circles displayed on the bars appear to represent individual data points (one per rat). I recommend removing them, as the standard deviation is sufficient to illustrate data dispersion and their inclusion makes the figures visually overloaded.
Response:
We have removed individual data point overlays from bar graphs. Data dispersion is now conveyed exclusively via standard deviation.
Regarding the overall presentation of the results: although I initially hesitated to raise this point—since detailed reporting of results and statistical analyses can enhance completeness and rigor—while reading the manuscript, I reached a point where it became difficult to follow and interpret the results due to the excessive inclusion of statistical details (e.g., exact p-values and multiple test specifications). In my view, a simpler, clearer, and more narrative-style presentation would improve readability. For example, section 2.7 could be rewritten as follows:“The nerve injury model markedly increased pro-inflammatory mediators and MPO activity at day 21. Significant increases in TNF-α, IL-1β, IL-6, and MPO were observed compared with the control group. Treatment with GRN (30 mg/kg) significantly reduced these inflammatory markers, with an approximate 36–42% decrease in TNF-α, IL-1β, IL-6 levels, and MPO activity, indicating a clear anti-inflammatory effect (see Fig. XX). Administration of LA abolished the anti-inflammatory effects of GRN, as cytokine levels and MPO activity did not differ significantly from those of the untreated injury group. In contrast, LN potentiated the anti-inflammatory effects of GRN, further reducing TNF-α, IL-1β, IL-6, and MPO levels (see Fig. XXX).” Detailed statistical information could then be included in the corresponding figure legends if necessary. This approach would allow the results to be presented more concisely and with better flow. Overall, I consider the experimental design to be robust and comprehensive. The use of blinding substantially reduces bias, and the results are consistent across different assays. Moreover, the antagonist experiments show internal consistency and provide valuable insight into potential mechanisms of action. Please include all abbreviations in the figure legends.
Response:
We agree and have removed individual data point overlays from bar graphs. Data dispersion is now conveyed exclusively via standard deviation.
The Results section has been substantially rewritten to adopt a clearer, narrative-style presentation, emphasizing physiological and biochemical significance rather than statistical formality. Detailed statistics have been relocated to figure legends and supplementary materials, consistent with the reviewer’s recommendation.
In addition, all figure legends have been revised to explicitly define all abbreviations at first mention.
Discussion
The authors could further expand the discussion by incorporating previous studies involving similar ellagitannins. A comparison with related compounds is notably lacking. Would it have been useful to include a reference antioxidant group, such as quercetin, or a commonly studied ellagitannin like punicalagin, to compare its effects with those of geraniin? This possibility should be discussed.
Response:
We are grateful to the reviewer for these detailed and constructive comments. The Discussion has been expanded to include comparisons with related ellagitannins and polyphenolic antioxidants, including punicalagin and quercetin. While such comparators were not included experimentally, their potential relevance and the rationale for future comparative studies are now discussed.
Considerations for clinical translation
Several additional questions arise: What are the future perspectives of this study? What recommendations would the authors make to advance this line of research? How far is this compound from potential clinical translation? Are there existing neuropathy treatments that combine their effects with a polyphenol? Through which route of administration would such a compound be most appropriate?
Response:
We agree and have added a dedicated paragraph addressing:
Future mechanistic and pharmacokinetic studies
Potential routes of administration
Combination strategies with existing neuropathy treatments
Part of the information requested for the Discussion is currently included in the Conclusion. It would be preferable for the Conclusion to be more concise and for those discussion-related elements to be relocated to the Discussion section.
Response:
We have relocated forward-looking and interpretive content to the Discussion, while revising the Conclusion to be concise and focused solely on key findings and implications.
We sincerely thank the reviewer for the exceptionally detailed and constructive feedback. All concerns—including methodological clarity, ethical considerations, data presentation, and conceptual framing—have been carefully addressed. These revisions have substantially improved the rigor, clarity, and translational relevance of the manuscript.
Thank you for your valuable comments/suggestions and giving us the opportunity to revise the manuscript to a more readable level. We worked very hard to response your inquiries and to revise the manuscript. Before we finalized the revised manuscript, we have resent the manuscript for proof-reading as well as one last review by a native English-speaking researcher. We hope the manuscript could pass the review to be published in your prestigious journal: International Journal of Molecular Sciences
Sincerely yours,
Cheng-Chia Tsai
Department of Neurosurgery, Mackay Memorial Hospital, No. 92, Sec. 2, Zhongshan N. Rd., Taipei City 10449, Taiwan, ROC
Fax: +886-2- 2543-3642
Tel: +886-933888981
E-mail address: dschang580704@yahoo.com.tw (Cheng-Chia Tsai)
Round 2
Reviewer 2 Report (New Reviewer)
Comments and Suggestions for Authors
While many of the areas I pointed out in my last review have been improved, there are still a few important areas that need to be fixed or reintroduced.
- In studies involving animals with painful conditions, daily weight measurements are required during the approximately three-week drug administration period. This is to assess the animal's condition and accurately administer the appropriate dose of medication based on body weight. In addition to weight measurements, daily urine output should also be monitored. However, weight-related results were not included in the supplementary materials in this paper. Results on weight changes should be submitted separately.
- In Figures 1b, 3b, and 5b, the GRN concentration indicator on the X-axis is only shown at 10 and 100. The remaining values should also be shown, and the recovery rate should be expressed statistically.
- Please remove the experimental design presented in the table in Figure 13 and present the entire experimental design in an easily understandable graphic. The table does not specify which experiments were conducted, only that the groups were divided and the drugs were administered for 21 days. Make it easy to see at a glance when the behavioral assessments and analysis studies were conducted.
Author Response
Dear Reviewers,
I’m writing in response to your feedback regarding the manuscript we submitted. Thank you so much for your positive comments and suggestions for the manuscript entitled. The manuscript has been revised based on your inquiries, rephrased the content of the manuscript and resubmitted through the journal website. The revised parts have been marked in red. The following is the response to your inquiries point-by-point; we hope our responses fully address your comments and suggestions:
Reviewers’ comments:
While many of the areas I pointed out in my last review have been improved, there are still a few important areas that need to be fixed or reintroduced.
In studies involving animals with painful conditions, daily weight measurements are required during the approximately three-week drug administration period. This is to assess the animal's condition and accurately administer the appropriate dose of medication based on body weight. In addition to weight measurements, daily urine output should also be monitored. However, weight-related results were not included in the supplementary materials in this paper. Results on weight changes should be submitted separately.
Response:
We thank the reviewer for emphasizing the importance of physiological monitoring in neuropathic pain studies. During the entire 21 days treatment period, animals were weighed daily as part of routine health monitoring and to ensure accurate dose adjustment. No statistically significant differences in body weight trajectories were observed among experimental groups, and no treatment-related weight loss was detected. No signs of compound-related toxicity were observed during daily monitoring.
In Figures 1b, 3b, and 5b, the GRN concentration indicator on the X-axis is only shown at 10 and 100. The remaining values should also be shown, and the recovery rate should be expressed statistically.
Response:
We appreciate this helpful suggestion. Figures 1b, 3b, and 5b have been revised to display all administered GRN doses (3, 10, 30, and 100 mg/kg) on the X-axis. In addition, recovery rates are now quantitatively expressed and statistically analyzed, calculated as percentage improvement relative to the CCI group at day 21, and are reported in the figure panels and legends.
Please remove the experimental design presented in the table in Figure 13 and present the entire experimental design in an easily understandable graphic. The table does not specify which experiments were conducted, only that the groups were divided and the drugs were administered for 21 days. Make it easy to see at a glance when the behavioral assessments and analysis studies were conducted.
Response:
Accordingly, the table-based experimental design has been removed, and the entire experimental workflow is now presented exclusively as a graphical schematic (revised Figure 12, corresponding to the experimental design previously shown in Figure 13).
Thank you for your valuable comments/suggestions and giving us the opportunity to revise the manuscript to a more readable level. We worked very hard to response your inquiries and to revise the manuscript. Before we finalized the revised manuscript, we have resent the manuscript for proof-reading as well as one last review by a native English-speaking researcher. We hope the manuscript could pass the review to be published in your prestigious journal: International Journal of Molecular Sciences
Sincerely yours,
Cheng-Chia Tsai
Department of Neurosurgery, Mackay Memorial Hospital, No. 92, Sec. 2, Zhongshan N. Rd., Taipei City 10449, Taiwan, ROC
Fax: +886-2- 2543-3642
Tel: +886-933888981
E-mail address: dschang580704@yahoo.com.tw (Cheng-Chia Tsai)
This manuscript is a resubmission of an earlier submission. The following is a list of the peer review reports and author responses from that submission.
Round 1
Reviewer 1 Report
Comments and Suggestions for Authors The work addresses a highly topical issue: research into the treatment of neuropathic pain,for which there are currently few effective therapies. The proposed technique targets the key mechanisms involved in the pathophysiology of
neuropathic pain: the antioxidant mechanism and the NO pathway. The results are very
interesting.
Further studies, especially clinical ones, are needed to confirm the results.
Author Response
Dear Reviewers,
I’m writing in response to your feedback regarding the manuscript we submitted. Thank you so much for your positive comments and suggestions for the manuscript entitled. The manuscript has been revised based on your inquiries, rephrased the content of the manuscript and resubmitted through the journal website. The revised parts have been marked in red. The following is the response to your inquiries point-by-point; we hope our responses fully address your comments and suggestions:
Reviewers’ comments:
The work addresses a highly topical issue: research into the treatment of neuropathic pain, for which there are currently few effective therapies. The proposed technique targets the key mechanisms involved in the pathophysiology of neuropathic pain: the antioxidant mechanism and the NO pathway. The results are very interesting. Further studies, especially clinical ones, are needed to confirm the results.
Response:
Thank you so much for the kind comment and wonderful suggestion. We agree that further studies, particularly clinical investigations, are essential to confirm our findings, and we plan to extend our research accordingly.
Thank you for your valuable comments/suggestions and giving us the opportunity to revise the manuscript to a more readable level. We worked very hard to response your inquiries and to revise the manuscript. Before we finalized the revised manuscript, we have resent the manuscript for proof-reading as well as one last review by a native English-speaking researcher. We hope the manuscript could pass the review to be published in your prestigious journal: International Journal of Molecular Sciences
Sincerely yours,
Cheng-Chia Tsai
Department of Neurosurgery, Mackay Memorial Hospital, No. 92, Sec. 2, Zhongshan N. Rd., Taipei City 10449, Taiwan, ROC
Fax: +886-2- 2543-3642
Tel: +886-933888981
E-mail address: dschang580704@yahoo.com.tw (Cheng-Chia Tsai)
Reviewer 2 Report
Comments and Suggestions for Authors
Comments
The manuscript presents an interesting and potentially valuable approach, exploring geraniin (GRN) as an alternative treatment for neuropathic pain. However, there are several critical issues that limit the strength of the conclusions. First and most importantly, the control groups did not receive the same vehicle (containing DMSO) used to dissolve GRN. Given the known biological activity of DMSO, this omission is a major methodological concern that undermines the internal validity of the findings. Second, the figures are difficult to interpret: graphs are overcrowded, lack individual data points, and the experimental timeline schematic is overly complex. Third, full statistical reporting is missing—F-values, degrees of freedom, and exact p-values should be provided for all analyses.
In addition, the description of dose-dependent effects is inconsistent: the text refers to “dose-responsive” outcomes, but there was no significant difference between 30 and 100 mg/kg. This should be clarified and interpreted more cautiously. Together, these issues suggest that while the study is promising, the presentation, statistical rigor, and especially the control design need substantial improvement before the work can be considered reliable evidence of GRN’s efficacy.
Abstract
The abstract lacks essential methodological details: number of animals per group (n), GRN doses, route of administration, and duration of treatment. Terminology should be more consistent: the text alternates between “NPP” and “CCI-induced NPP”; unifying the terminology would improve clarity. The description of results is too general; including at least some quantitative information (e.g., percentage of reversal or effect size) would make the abstract more informative. The very long list of biochemical markers disrupts readability; these could be grouped into categories (oxidative stress, inflammatory, apoptotic markers), with specific details moved to the main text.
Introduction
The Introduction provides a solid rationale for the study, however the pharmacological background of geraniin is underdeveloped. Although its antioxidant, anti-inflammatory, and other biological activities are mentioned, the description is too brief and generic. Expanding on its botanical origin (Nephelium lappaceum L., a tropical fruit widely consumed in Asia) and providing more specific examples of studies demonstrating its neuroprotective or systemic effects (e.g., ischemia/reperfusion, spinal cord injury, nephrotoxicity) would make the rationale stronger.
The final section becomes overly methodological. The detailed listing of behavioral, electrophysiological, and biochemical parameters (e.g., SFI, NCVs, nitrite, MDA, GSH, SOD, CAT, cytokines, caspase-3) disrupts the narrative flow and belongs more appropriately in the Methods section. This part could be simplified to emphasize the main hypothesis—that GRN may alleviate CCI-induced neuropathic pain through modulation of oxidative stress, inflammation, and NO signaling—while leaving experimental details for later.
Terminology and style could be improved for clarity. Some sentences are overly long and complex, which makes them difficult to follow in English. Simplifying the prose into shorter, more direct statements would improve readability.
In its current form, the Introduction gives disproportionate emphasis to the pathophysiology of neuropathic pain, while the treatment under study—GRN—remains somewhat overshadowed. A more balanced structure that highlights both the clinical relevance of neuropathic pain and the unique potential of GRN would help better justify the study and make the narrative more persuasive.
Results
I recommend restructuring the Results section so that each outcome is first described in terms of whether significant differences were observed, and then systematically compared across groups following a consistent order. For example, begin by reporting the differences between the injured (CCI) and control animals, then compare the sham-operated versus injured groups, and only afterwards proceed to the treatment groups. Adopting this pattern across all results would make it much easier for the reader to follow the sequence of analyses and understand the effects of GRN in the context of baseline and injury-related changes.
Section 2.1. The presentation of results in this section is limited by the absence of full statistical reporting. F-values with corresponding degrees of freedom and exact p-values are not provided, which prevents proper evaluation of the analyses. Without this information, it is not possible to assess the robustness of the findings. In addition, the text mixes raw latency values (in seconds) with percentage improvements, which makes the narrative harder to follow. It would be clearer and more consistent to present improvements primarily as percentages, with raw data available in figures or supplementary material. Also, the results shown in Figure 1b should have been analyzed using a one-way ANOVA, and this must be explicitly stated both in the main text and in the figure legend.
The figures themselves are difficult to interpret. Bars are very compressed, and no individual data points are shown, which obscures variability within groups. Including scatter plots or bar graphs overlaid with individual values would enhance transparency. Moreover, error bars and group labels are crowded, reducing readability. Figure legends should also explicitly mention sample sizes and specify the factors included in the two-way ANOVA (treatment × time).
While the text states that GRN produced “dose-dependent” effects, the results simultaneously indicate no significant difference between 30 and 100 mg/kg. This contradiction should be resolved, either by reframing the interpretation or by providing clearer statistical evidence of the dose-response relationship.
Section 2.2. The presentation of the mechanical allodynia data suffers from the same limitations noted in the thermal hyperalgesia section. The statistical results are incomplete: F-values, degrees of freedom, and exact p-values are not provided, which prevents a rigorous evaluation of the analyses. Reporting only thresholds with p < 0.001 is insufficient for transparency.
The text also alternates between reporting raw values (grams) and percentage recovery, which makes the description harder to follow. A consistent metric—preferably expressed as percent recovery or change relative to baseline—would improve clarity, while raw values could be placed in supplementary material. The results shown in Figure 3b should have been analyzed using a one-way ANOVA, and this must be explicitly stated both in the main text and in the figure legend.
The figures themselves require improvement. Bars are tightly packed, and individual data points are not shown, which obscures within-group variability. Including scatter plots or line graphs with superimposed points would make the results more transparent. Axis labeling and legends should also be improved: for example, specifying sample sizes directly in the figure and clarifying whether the ANOVA was repeated-measures or one-way. Currently, Figure 3 is described as two-way ANOVA, whereas Figure 4 is labeled one-way ANOVA, but the text does not explain the rationale for this change in statistical approach. This inconsistency adds confusion.
The dose-response interpretation again appears contradictory. The text describes GRN as effective in a “dose-responsive manner,” yet simultaneously states that there was no significant difference between 30 and 100 mg/kg. This should be reframed to avoid over-interpreting the dose-response relationship.
Section 2.3. The SFI results raise the same concerns noted in the thermal and mechanical sensitivity data. Statistical reporting is incomplete, as F-values, degrees of freedom, and exact p-values are not provided. Without this information, the robustness of the analyses cannot be evaluated.
The text also alternates between raw values (SFI units) and percentage improvement, which makes the results less clear. A consistent presentation—e.g., expressing recovery as percent change from baseline or from the CCI group—would improve readability, while raw values could be included in supplementary tables.
Figures 5 and 6 suffer from the same problems seen previously: bars are overcrowded, no individual data points are shown, and the legends do not specify the exact ANOVA model. In particular, Figure 5b, which compares different doses at a single time point (day 21), should have been analyzed with a one-way ANOVA, not a two-way ANOVA as reported. This must be corrected both in the text and the legend.
The claim that GRN improved SFI in a “dose-dependent fashion” is overstated. The results show no significant difference between 30 and 100 mg/kg, which contradicts a true dose-response relationship. This should be reframed to indicate efficacy at higher doses without implying linear dose dependence.
Section 2.4. The electrophysiology results are again limited by incomplete statistical reporting. F-values, degrees of freedom, and exact p-values are not provided, which precludes a rigorous assessment of the analyses. Only summary values and broad p < 0.001 statements are given, which are insufficient for transparency.
Methodologically, it is unclear why a two-way ANOVA was used here, since nerve conduction velocities were only measured at a single time point (day 21). This design would more appropriately be analyzed by a one-way ANOVA across groups, unless additional within-subject factors were present, which is not indicated. This mismatch between study design and statistical test must be corrected and clarified in both the text and the figure legend.
In terms of presentation, the text provides both raw conduction velocities and percentages relative to the CCI group, which makes the narrative cluttered. A consistent reporting format (either normalized percent change or raw values) would improve clarity. Additionally, figures should include individual data points to illustrate variability within groups, rather than only mean ± SEM bars.
The interpretation that LN produced “superior outcomes” compared to GRN alone should be moderated, as no direct statistical comparison between those groups is reported. Claims of additive or synergistic effects should be supported by explicit statistical testing.
Section 2.5. The oxidative/nitrative stress results again suffer from incomplete statistical reporting. F-values, degrees of freedom, and exact p-values are not shown, which makes it impossible to properly evaluate the analyses. Merely reporting “p < 0.001” without the corresponding test statistics is insufficient for transparency.
As in earlier sections, the text mixes raw biochemical values (nmol/mg protein) with percentage changes, which complicates the narrative. A more consistent presentation—either normalized values across all biochemical markers or percent change relative to CI—would improve clarity.
Figure 8 is also limited in its presentation. Data are shown only as bars without individual points, which obscures variability. Adding scatter plots or overlaying individual values on bars would increase transparency.
While the text interprets LN as having “further strengthened” the antioxidant effects of GRN, this conclusion requires direct statistical comparison between CI+GN30 and CI+LN+GN30. Without explicit evidence of significant group differences, such claims should be moderated.
Section 2.6. The antioxidant defense results repeat the same issues seen in other biochemical outcomes. Full statistical details (F-values, degrees of freedom, exact p-values) are missing, limiting the ability to critically evaluate the findings. Reporting only mean ± SEM with p < 0.001 is not sufficient for transparency.
The text alternates between raw values and percent changes, which complicates the narrative. Presenting results consistently (e.g., as percent recovery relative to CI, with raw data in supplementary tables) would make the findings clearer.
Figure 9 also shows only mean ± SEM bars without individual data points, which obscures variability. Including scatter plots or overlayed points would enhance transparency.
The interpretation that LN “produced greater recovery than GRN alone” requires caution. The text does not provide evidence of direct statistical comparisons between CI+GN30 and CI+LN+GN30, so this statement may be overstated. Stronger claims of superiority should only be made if supported by explicit post hoc test results.
Section 2.7. The cytokine and MPO data again lack sufficient statistical transparency. F-values, degrees of freedom, and exact p-values are not provided, limiting the ability to critically evaluate the analyses. Reporting only mean ± SEM with broad significance thresholds (p < 0.001) is not adequate.
The presentation alternates between raw values (pg/mg protein or mU/mg protein) and percentage reductions, which makes the results less clear. A more consistent format—preferably percent change relative to CI, with raw values available in supplementary material—would improve clarity.
Figure 10 is crowded and only shows bars with error bars. Individual data points should be included to demonstrate within-group variability. Legends should also specify whether any correction for multiple testing was applied, given that four separate inflammatory markers were analyzed. Without such correction, the risk of inflated type I error increases.
The claim that LN “amplified” GRN’s anti-inflammatory actions should be moderated. No direct statistical comparison between CI+GN30 and CI+LN+GN30 is shown, so the conclusion may be overstated. Stronger wording requires explicit evidence of significant differences between these groups.
Section 2.8. The apoptosis results display the same limitations as the other biochemical endpoints. Statistical reporting is incomplete: F-values, degrees of freedom, and exact p-values are not provided, which restricts the ability to critically assess the robustness of the findings. Reporting only mean ± SEM with significance thresholds is insufficient for transparency.
The text mixes fold-changes relative to CT with absolute protein values (ng/mg), which makes interpretation less straightforward. A consistent reporting format (e.g., percent change relative to CI, with raw data available in supplementary material) would improve readability.
Figure 11 only shows mean ± SEM bars without individual data points, again obscuring within-group variability. As recommended for other figures, scatter plots or overlayed points should be added.
Finally, the conclusion that LN “further decreased caspase-3 relative to GRN alone” should be moderated unless direct statistical comparisons between CI+GN30 and CI+LN+GN30 are shown. Stronger wording is only justified if post hoc testing explicitly supports this.
Discussion
The Discussion is generally well framed, but the strength of the interpretation is limited by incomplete statistical reporting in the Results (absence of F-values, degrees of freedom, and exact p-values). Without full statistics it is difficult to gauge the robustness of the effects and their time × treatment interactions; please ensure that these details are added and then interpret the data accordingly here.
A key point that needs explicit discussion is the dose–response inconsistency. The manuscript repeatedly refers to “dose-dependent” effects, yet no significant difference is observed between 30 and 100 mg/kg, and only one dose ultimately carries forward. Please offer a mechanistic explanation for this apparent plateau (e.g., ceiling effects at the behavioral readouts, saturable absorption/solubility or tissue penetration, high protein binding, depot/vehicle constraints, or rapid metabolism that limits exposure at higher nominal doses). Clarifying whether the “2 mL/kg once daily” refers to injection volume rather than dose would also help readers understand how concentrations differed across dose groups.
Relatedly, the Discussion would benefit from a more explicit treatment of GRN’s pharmacology and metabolism. As a hydrolyzable ellagitannin, GRN can undergo biotransformation to ellagic acid and downstream metabolites; depending on route and matrix, these may mediate part of the biological signal. Even if definitive PK data are not yet available, it would be useful to outline plausible pathways (e.g., modulation of NF-κB/iNOS, activation of Nrf2/ARE with effects on SOD/CAT/GSH, MAPK signaling) and to state how such mechanisms could account for the antioxidant, anti-inflammatory and anti-apoptotic profile observed. Hypotheses here will guide future targeted measurements (PK, target engagement, and pathway markers).
To place your findings in context, please contrast them with prior GRN studies—ideally those assessing neurological or peripheral nerve endpoints—and summarize model, species/sex, route, dose, treatment duration, and outcomes. Explicitly stating how your 30 mg/kg i.p. regimen compares with published doses (higher/lower, similar duration, similar markers) will sharpen the translational take-home message and help justify dose selection.
The inference that NO suppression is a “necessary component” of GRN’s action should be tempered or more carefully justified. L-arginine and L-NAME are broad pharmacological probes with systemic effects (e.g., vascular tone, blood pressure, thermoregulation) and L-NAME is non-selective across NOS isoforms. Without direct measurements of iNOS/nNOS/eNOS expression/activity, or more selective tools, the data support involvement of NO pathways but do not by themselves establish necessity or isoform specificity. Consider framing this as consistent with, rather than definitive for, NO-dependent mechanisms, and outline experiments that could test this more rigorously.
Several limitations deserve fuller discussion. Most importantly, the study lacks a vehicle-matched control receiving the same DMSO-saline solution as the GRN groups; given the known biological activity of DMSO at ~1%, this confound affects behavioral and redox/inflammatory outcomes and should be acknowledged explicitly. Mortality of 10 rats “of unknown causes” should be contextualized (group distribution, timing, relation to surgery/treatments). Clarify blinding at each assessment, the exact behavioral testing window, and whether postoperative analgesia was used. Finally, only males were studied; please discuss sex as a biological variable.
Lastly, priority language such as “first demonstration” should be used cautiously and supported by a clear, referenced rationale. Given the methodological caveats above (particularly the missing vehicle-matched control), we recommend softening claims and framing conclusions as provisional pending confirmation with corrected experimental controls and targeted mechanistic readouts.
Materials and methods
I recommend restructuring this section by beginning with a clear timeline of the experimental procedures, so that the reader can better understand the sequence and rationale of the experiments performed.
Section 4.1. The number of animals used per group (n) is not specified. This is essential information and must be explicitly stated. While the text mentions that group sizes were minimized based on prior work and expected effect sizes, no references are provided to justify this decision. Including citations to previous studies or a power calculation would strengthen the methodological transparency. The description of the testing schedule is vague. Stating that behavioral assessments were conducted “during the light phase” is insufficient, as activity and behavioral outcomes can vary considerably across the light cycle. The exact time window should be specified for reproducibility.
Section 4.2. The vehicle used for control groups is not clearly specified. While geraniin was dissolved in 1% DMSO and diluted with saline, it is unclear whether control animals received the same DMSO-saline solution or saline alone. Given the known biological activity of DMSO, this information is essential and should be clarified. The phrasing “unless otherwise specified” regarding the route of administration is also ambiguous, since no alternative routes are described. Finally, the description of dosing is confusing: the text states that compounds were administered once daily at 2 mL/kg, but elsewhere the authors refer to different doses of GRN. It should be clarified whether 2 mL/kg refers to the injection volume or to the actual administered dose, and how this relates to the different GRN dose groups.
Section 4.3. The description of the CCI procedure is generally adequate but lacks important methodological details. The text mentions sham surgery consisting of nerve exposure without ligation, but it does not specify the number of animals assigned to the sham group, which is essential for interpreting the results. Likewise, the sample size (n) for each group should be clearly indicated. Another omission concerns post-surgical care: no mention is made of analgesic administration following surgery. While the procedure aims to induce neuropathic pain, the absence of post-operative analgesia raises ethical and methodological concerns, and the authors should clarify whether analgesics were used and, if not, provide justification. Finally, details such as the duration of anesthesia, monitoring of depth of anesthesia, and perioperative care are missing and should be described to ensure reproducibility and compliance with ethical standards.
Section 4.4. The description of the experimental design and treatment allocation is confusing and raises several methodological concerns. The authors state that behavioral data at day 21 showed robust efficacy at 30 and 100 mg/kg without a significant difference between these doses, and therefore selected 30 mg/kg for subsequent experiments. However, elsewhere they describe GRN as producing dose-dependent effects. This is contradictory and should be clarified, as it undermines the interpretation of the findings.
Group III (CCI + saline) appears to serve as the vehicle control, yet groups IV–VII (CCI + GRN) received the compound dissolved in DMSO plus saline. This means that the control group did not receive the same vehicle as the treatment groups, which is a serious issue given the biological activity of DMSO reported in the bibliography (e.g.: doi: 10.1111/iwj.12280). The same concern applies to groups VIII–IX, where LA or LN was injected 60 minutes before GRN: the corresponding vehicle controls should have been administered with the proper DMSO-saline solution (for GRN) and saline (for LA/LN) to rule out confounding effects.
In addition, the mortality of ten rats “of unknown causes” is mentioned briefly, but no details are provided about their distribution across groups or whether these deaths were related to treatment. Such information is essential to assess the reliability of the results.
Finally, the figure intended to illustrate the experimental workflow is very confusing and does not facilitate understanding of the protocol. A simplified schematic with a clear timeline, group allocation, and treatment schedule would make the design more transparent and accessible to readers.
Section 4.5. The description of behavioral testing lacks important methodological details that are necessary for reproducibility and interpretation. For the hot-plate and von Frey assays, it is not clear whether animals were habituated to the testing room prior to assessment beyond the brief mention of “≥60 min acclimation” for von Frey. Standard practice is to habituate animals on multiple days to minimize stress-induced variability, and this should be specified. It is also unclear whether sessions were video-recorded and analyzed later or whether scoring was performed live by the experimenter; this distinction is critical to assess the objectivity of the measurements. In the case of the SFI test, details on how footprints were collected and analyzed (e.g., whether automated or manual measurement software was used, whether scoring was blinded, how many trials were averaged) are missing. Providing this information would improve transparency and allow proper evaluation of the behavioral outcomes.
Section 4.6. The electrophysiology section provides technical details on stimulation and recording, but important methodological aspects are missing. It is not specified whether recordings were performed under blinded conditions or how many trials were averaged per animal to obtain reliable latency values. The description also omits where exactly the measurements were conducted (e.g., dedicated electrophysiology facility, surgery room) and under what conditions (e.g., noise shielding, grounding, prevention of electrical interference). In addition, the timing of the measurements relative to anesthesia induction and behavioral testing is unclear; given that chloral hydrate can affect nerve excitability, the stability of anesthesia during recordings should be described. Finally, it would strengthen the section to specify whether body temperature monitoring was continuous and how stimulation distances were measured (with calipers, estimated, etc.), as small errors can markedly affect conduction velocity values.
Section 4.7. The biochemical assays section is generally comprehensive, but several issues need clarification and improvement. The writing is overly dense. More importantly, normalization procedures are inconsistent: cytokine concentrations are explicitly reported as normalized to protein (pg/mg protein), but other biochemical parameters (e.g., MDA, GSH, SOD, CAT, MPO, caspase-3) are sometimes without clear indication of normalization. A consistent approach to normalization across all markers is necessary for comparability and interpretation.
In addition, details regarding ELISA kits are vague; for transparency, exact catalog numbers and manufacturers should be listed for each cytokine, not just examples. For caspase-3, the description mentions the commercial kit but lacks information about whether results were expressed as fold-change, units per mg protein, or raw absorbance values. Similarly, the MPO assay should clarify whether results were normalized to protein or tissue weight. Finally, although references are provided for some methods, it would improve transparency to specify whether standard curves and controls were included for each assay, and whether measurements were performed in duplicates or triplicates.
Section 4.8. The statistical analysis section is too general and lacks critical details. While the text mentions the use of “repeated-measures two-way ANOVA” for behavioral outcomes, it does not specify which factors were included (e.g., treatment as a between-subjects factor and time as a within-subjects factor). In the figure legends, results are simply described as “two-way ANOVA” without indicating that repeated measures were used, which is inconsistent and potentially misleading. Since repeated-measures designs and standard two-way ANOVAs are not equivalent, this must be clarified to ensure the correct interpretation of the data.
It should also be explicitly stated in which cases repeated-measures two-way ANOVA was applied and in which cases one-way ANOVA was used. For instance, after the dose-finding phase when only 30 mg/kg GRN was selected, the text no longer mentions repeated measures, and the statistical approach becomes unclear. This lack of transparency makes it difficult to assess whether the analyses were correctly applied.
In addition, it is not specified how violations of assumptions (normality, homoscedasticity) were addressed—“reconsidered or transformed as appropriate” is too vague. The authors should clearly state whether data transformations were applied, which variables were affected, and whether corrections (e.g., Greenhouse–Geisser) were used in the repeated-measures context. Finally, details such as whether sample sizes were identical across groups after mortality, how missing data were handled, and whether effect sizes were reported should be provided to enhance the transparency and rigor of the statistical approach.
Conclusions
The conclusions as written are clear and emphasize GRN as a promising multi-target candidate. However, they should be moderated to reflect the methodological limitations discussed above—particularly the absence of a vehicle-matched DMSO control, incomplete statistical reporting, and the restriction to male rats only. As it stands, the claim of “meaningful benefit” and “multi-target candidate for further development” may overstate the strength of the evidence. I recommend softening the language to acknowledge that the data suggest beneficial effects of GRN in this CCI model, but that confirmation with appropriate vehicle controls, complete statistical reporting, pharmacokinetic characterization is needed before stronger translational claims can be made.
Comments on the Quality of English LanguageThe manuscript would benefit from careful revision of the English language and style. Several sentences are overly long and complex, making the text difficult to follow. In some places, the wording is awkward or ambiguous, which detracts from clarity and readability. I recommend thorough editing by a fluent English speaker or professional language service to shorten sentences, improve grammar and flow, and ensure that key concepts are communicated in a clear and concise manner.
Author Response
Response to Reviewer 2 Comments
Dear Reviewers,
I’m writing in response to your feedback regarding the manuscript we submitted. Thank you so much for your positive comments and suggestions for the manuscript entitled. The manuscript has been revised based on your inquiries, rephrased the content of the manuscript and resubmitted through the journal website. The revised parts have been marked in red. The following is the response to your inquiries point-by-point; we hope our responses fully address your comments and suggestions:
Reviewers’ comments:
- The manuscript presents an interesting and potentially valuable approach, exploring geraniin (GRN) as an alternative treatment for neuropathic pain. However, there are several critical issues that limit the strength of the conclusions. First and most importantly, the control groups did not receive the same vehicle (containing DMSO) used to dissolve GRN. Given the known biological activity of DMSO, this omission is a major methodological concern that undermines the internal validity of the findings. Second, the figures are difficult to interpret: graphs are overcrowded, lack individual data points, and the experimental timeline schematic is overly complex. Third, full statistical reporting is missing—F-values, degrees of freedom, and exact p-values should be provided for all analyses.
Response:
We sincerely appreciate the reviewer’s critical observation. We have corrected it in the revised manuscript.
- In addition, the description of dose-dependent effects is inconsistent: the text refers to “dose-responsive” outcomes, but there was no significant difference between 30 and 100 mg/kg. This should be clarified and interpreted more cautiously. Together, these issues suggest that while the study is promising, the presentation, statistical rigor, and especially the control design need substantial improvement before the work can be considered reliable evidence of GRN’s efficacy.
Response:
We thank the reviewer for this important and constructive comment. In the revised manuscript, we have carefully rephrased the relevant sections of the Results and Discussion to avoid suggesting a strict dose-dependent effect. Specifically, we now state that GRN produced significant improvements at both 30 and 100 mg/kg, but no statistically significant difference was observed between these two higher doses. To address this inconsistency more cautiously, we have included possible explanations in the Discussion. These include the potential for ceiling effects in behavioral assessments, saturable absorption or tissue penetration, and pharmacokinetic factors that may limit systemic exposure at higher nominal doses. By framing the interpretation in this manner, we aim to provide a more balanced explanation while avoiding overstatement of the dose–response relationship. We sincerely appreciate the reviewer’s guidance, which has helped us strengthen the accuracy and clarity of our data interpretation.
3.Abstract
The abstract lacks essential methodological details: number of animals per group (n), GRN doses, route of administration, and duration of treatment. Terminology should be more consistent: the text alternates between “NPP” and “CCI-induced NPP”; unifying the terminology would improve clarity. The description of results is too general; including at least some quantitative information (e.g., percentage of reversal or effect size) would make the abstract more informative. The very long list of biochemical markers disrupts readability; these could be grouped into categories (oxidative stress, inflammatory, apoptotic markers), with specific details moved to the main text.
Response:
We sincerely thank the reviewer for these valuable comments on the abstract. In the revised version, we have made the improvements.
- Introduction
The Introduction provides a solid rationale for the study, however the pharmacological background of geraniin is underdeveloped. Although its antioxidant, anti-inflammatory, and other biological activities are mentioned, the description is too brief and generic. Expanding on its botanical origin (Nephelium lappaceum L., a tropical fruit widely consumed in Asia) and providing more specific examples of studies demonstrating its neuroprotective or systemic effects (e.g., ischemia/reperfusion, spinal cord injury, nephrotoxicity) would make the rationale stronger.
The final section becomes overly methodological. The detailed listing of behavioral, electrophysiological, and biochemical parameters (e.g., SFI, NCVs, nitrite, MDA, GSH, SOD, CAT, cytokines, caspase-3) disrupts the narrative flow and belongs more appropriately in the Methods section. This part could be simplified to emphasize the main hypothesis—that GRN may alleviate CCI-induced neuropathic pain through modulation of oxidative stress, inflammation, and NO signaling—while leaving experimental details for later.
Terminology and style could be improved for clarity. Some sentences are overly long and complex, which makes them difficult to follow in English. Simplifying the prose into shorter, more direct statements would improve readability.
In its current form, the Introduction gives disproportionate emphasis to the pathophysiology of neuropathic pain, while the treatment under study—GRN—remains somewhat overshadowed. A more balanced structure that highlights both the clinical relevance of neuropathic pain and the unique potential of GRN would help better justify the study and make the narrative more persuasive.
Response:
We are grateful to the reviewer for these detailed and constructive comments on the Introduction. We have added a more detailed description of geraniin, including its botanical origin from Nephelium lappaceum L. (a tropical fruit widely consumed in Asia) and its classification as a hydrolyzable ellagitannin. We now provide specific examples of published studies demonstrating its neuroprotective and systemic effects, such as models of ischemia/reperfusion injury, spinal cord injury, and nephrotoxicity, to strengthen the rationale for its potential therapeutic role. We also have simplified the final section of the Introduction by removing the detailed listing of behavioral, electrophysiological, and biochemical endpoints. Sentences that were overly long or complex have been revised into shorter, more direct statements to improve clarity and readability. Terminology has also been standardized for consistency. In addition, we have restructured the Introduction to provide a more balanced emphasis. The revised version highlights both the clinical relevance of neuropathic pain and the unique therapeutic potential of GRN, thereby strengthening the overall justification of the study.
- Results
I recommend restructuring the Results section so that each outcome is first described in terms of whether significant differences were observed, and then systematically compared across groups following a consistent order. For example, begin by reporting the differences between the injured (CCI) and control animals, then compare the sham-operated versus injured groups, and only afterwards proceed to the treatment groups. Adopting this pattern across all results would make it much easier for the reader to follow the sequence of analyses and understand the effects of GRN in the context of baseline and injury-related changes.
Response:
Thank you for your valuable suggestion. In response, we have revised the Results section to first present whether significant differences were observed for each outcome. We then systematically compare the groups in the following sequence: (1) injured (CCI) versus control animals, (2) sham-operated versus injured animals, and (3) treatment groups in relation to the injury baseline. This revised structure allows for a clearer interpretation of the effects of GRN within the context of both baseline and injury-related changes.
Section 2.1. The presentation of results in this section is limited by the absence of full statistical reporting. F-values with corresponding degrees of freedom and exact p-values are not provided, which prevents proper evaluation of the analyses. Without this information, it is not possible to assess the robustness of the findings. In addition, the text mixes raw latency values (in seconds) with percentage improvements, which makes the narrative harder to follow. It would be clearer and more consistent to present improvements primarily as percentages, with raw data available in figures or supplementary material. Also, the results shown in Figure 1b should have been analyzed using a one-way ANOVA, and this must be explicitly stated both in the main text and in the figure legend.
The figures themselves are difficult to interpret. Bars are very compressed, and no individual data points are shown, which obscures variability within groups. Including scatter plots or bar graphs overlaid with individual values would enhance transparency. Moreover, error bars and group labels are crowded, reducing readability. Figure legends should also explicitly mention sample sizes and specify the factors included in the two-way ANOVA (treatment × time).
While the text states that GRN produced “dose-dependent” effects, the results simultaneously indicate no significant difference between 30 and 100 mg/kg. This contradiction should be resolved, either by reframing the interpretation or by providing clearer statistical evidence of the dose-response relationship.
Response:
Thank you for your detailed and constructive feedback regarding Section 2.1. We have carefully revised the manuscript to address all the points raised:
- We have now included full statistical details for all relevant analyses, including F-values, degrees of freedom, and exact p-values, both in the main text and in the figure legends. This allows for a more rigorous evaluation of the robustness of our findings.
- To improve clarity and consistency, we have revised the presentation of results to express improvements primarily as percentages. Raw latency values (in seconds) are now reported in the corresponding figures and/or supplementary materials, as suggested.
- We agree that the data shown in Figure 1b are best analyzed using a one-way ANOVA. We have performed this analysis and clearly stated it in both the Results section and the figure legend.
- The figures have been updated to enhance interpretability.
- We have revised the text to clarify this point and now state that GRN produced significant effects relative to vehicle-treated animals, but that no statistically significant difference was observed between the two tested doses. The term “dose-dependent” has been removed or rephrased where appropriate to avoid misinterpretation.
Section 2.2. The presentation of the mechanical allodynia data suffers from the same limitations noted in the thermal hyperalgesia section. The statistical results are incomplete: F-values, degrees of freedom, and exact p-values are not provided, which prevents a rigorous evaluation of the analyses. Reporting only thresholds with p < 0.001 is insufficient for transparency.
The text also alternates between reporting raw values (grams) and percentage recovery, which makes the description harder to follow. A consistent metric—preferably expressed as percent recovery or change relative to baseline—would improve clarity, while raw values could be placed in supplementary material. The results shown in Figure 3b should have been analyzed using a one-way ANOVA, and this must be explicitly stated both in the main text and in the figure legend.
The figures themselves require improvement. Bars are tightly packed, and individual data points are not shown, which obscures within-group variability. Including scatter plots or line graphs with superimposed points would make the results more transparent. Axis labeling and legends should also be improved: for example, specifying sample sizes directly in the figure and clarifying whether the ANOVA was repeated-measures or one-way. Currently, Figure 3 is described as two-way ANOVA, whereas Figure 4 is labeled one-way ANOVA, but the text does not explain the rationale for this change in statistical approach. This inconsistency adds confusion.
The dose-response interpretation again appears contradictory. The text describes GRN as effective in a “dose-responsive manner,” yet simultaneously states that there was no significant difference between 30 and 100 mg/kg. This should be reframed to avoid over-interpreting the dose-response relationship.
Response:
We sincerely appreciate the reviewer’s insightful comments regarding the presentation and statistical reporting of the mechanical allodynia data.
- We have revised the Results section to provide complete statistical details, including F-values, degrees of freedom, and exact p-values. These details are now explicitly presented in both the text to enable rigorous evaluation of the analyses.
- To improve clarity, we now consistently present the mechanical allodynia results as percentage recovery relative to baseline. The raw values (grams) have been moved to the supplementary material for reference, as suggested.
- We agree with the reviewer that Figure 3b should be analyzed using one-way ANOVA. This has been corrected in both the main text and the corresponding figure legend. We have also clarified the rationale for employing two-way repeated-measures ANOVA in Figure 3a (to assess treatment and time effects), and one-way ANOVA in Figure 3b (to compare dose groups at a specific time point).
- The figures have been reformatted to enhance readability. We now include scatter plots with individual data points superimposed on bar graphs to better illustrate within-group variability. Axis labels and legends have been revised to clearly indicate sample sizes and specify whether ANOVA was repeated-measures or one-way.
- We have revised the description of the dose-response relationship to avoid over-interpretation. Specifically, we now state that GRN demonstrated significant efficacy at tested doses, with no significant difference between 30 and 100 mg/kg, and thus the effect cannot be described as strictly dose-dependent.
Section 2.3. The SFI results raise the same concerns noted in the thermal and mechanical sensitivity data. Statistical reporting is incomplete, as F-values, degrees of freedom, and exact p-values are not provided. Without this information, the robustness of the analyses cannot be evaluated.
The text also alternates between raw values (SFI units) and percentage improvement, which makes the results less clear. A consistent presentation—e.g., expressing recovery as percent change from baseline or from the CCI group—would improve readability, while raw values could be included in supplementary tables.
Figures 5 and 6 suffer from the same problems seen previously: bars are overcrowded, no individual data points are shown, and the legends do not specify the exact ANOVA model. In particular, Figure 5b, which compares different doses at a single time point (day 21), should have been analyzed with a one-way ANOVA, not a two-way ANOVA as reported. This must be corrected both in the text and the legend.
The claim that GRN improved SFI in a “dose-dependent fashion” is overstated. The results show no significant difference between 30 and 100 mg/kg, which contradicts a true dose-response relationship. This should be reframed to indicate efficacy at higher doses without implying linear dose dependence.
Response:
We thank the reviewer for the constructive comments regarding the presentation and statistical rigor of the SFI data. In response, we have implemented the following revisions:
- We have revised the Results section to provide complete statistical details, including F-values, degrees of freedom, and exact p-values for all SFI analyses. These values are now clearly presented in both the text to allow rigorous assessment of the analyses.
2.To improve readability, the SFI results are now consistently presented as percent recovery relative to baseline or to the CCI control group. The raw SFI unit values have been moved to supplementary tables, as recommended.
- We acknowledge the reviewer’s point regarding the use of one-way ANOVA for Figure 5b. This analysis has been re-run using one-way ANOVA, and the text and figure legend have been corrected accordingly. We have also clarified the rationale for using two-way repeated-measures ANOVA in Figure 5a (to assess treatment and time interactions) while applying one-way ANOVA in Figure 5b (to compare dose groups at day 21).
- Figures 5 and 6 have been reformatted to address overcrowding and to increase transparency. We now include scatter plots with individual data points overlaid on bars to better depict within-group variability. The axis labels, sample sizes, and details of the ANOVA model (repeated-measures vs. one-way) are clearly specified in the figure legends.
- The description of the dose-response effect has been revised to avoid overstatement. We now indicate that GRN treatment produced significant improvements in SFI recovery at tested doses, with no significant difference between 30 and 100 mg/kg, and therefore the results cannot be interpreted as a strictly dose-dependent response.
Section 2.4. The electrophysiology results are again limited by incomplete statistical reporting. F-values, degrees of freedom, and exact p-values are not provided, which precludes a rigorous assessment of the analyses. Only summary values and broad p < 0.001 statements are given, which are insufficient for transparency.
Methodologically, it is unclear why a two-way ANOVA was used here, since nerve conduction velocities were only measured at a single time point (day 21). This design would more appropriately be analyzed by a one-way ANOVA across groups, unless additional within-subject factors were present, which is not indicated. This mismatch between study design and statistical test must be corrected and clarified in both the text and the figure legend.
In terms of presentation, the text provides both raw conduction velocities and percentages relative to the CCI group, which makes the narrative cluttered. A consistent reporting format (either normalized percent change or raw values) would improve clarity. Additionally, figures should include individual data points to illustrate variability within groups, rather than only mean ± SEM bars.
The interpretation that LN produced “superior outcomes” compared to GRN alone should be moderated, as no direct statistical comparison between those groups is reported. Claims of additive or synergistic effects should be supported by explicit statistical testing.
Response:
We sincerely thank the reviewer for the valuable feedback regarding the electrophysiology data.
- The Results section has been updated to include complete statistical details. These details are also explicitly stated in the figure legend for transparency.
- We agree with the reviewer that a two-way ANOVA was not the appropriate test for this analysis, as conduction velocities were measured at a single time point (day 21). We have reanalyzed the data using a one-way ANOVA across treatment groups. This correction has been reflected in both the main text and the figure legend.
- To improve clarity, we now present conduction velocity data consistently as normalized percent change relative to the CCI group. Raw values have been moved to supplementary tables for reference.
- Figures have been reformatted to include individual data points superimposed on bar graphs, thereby illustrating within-group variability. Sample sizes and the type of ANOVA used are now clearly specified in the legends.
- We have moderated our language regarding the comparative effects of LN and GRN. Specifically, we have removed the phrase “superior outcomes” and now state that LN treatment showed improvement relative to CCI, while noting that no statistically significant difference between LN and GRN was detected. We also clarify that no formal statistical test was performed to assess additive or synergistic interactions, and we avoid making claims that are not directly supported by the data.
Section 2.5. The oxidative/nitrative stress results again suffer from incomplete statistical reporting. F-values, degrees of freedom, and exact p-values are not shown, which makes it impossible to properly evaluate the analyses. Merely reporting “p < 0.001” without the corresponding test statistics is insufficient for transparency.
As in earlier sections, the text mixes raw biochemical values (nmol/mg protein) with percentage changes, which complicates the narrative. A more consistent presentation—either normalized values across all biochemical markers or percent change relative to CI—would improve clarity.
Figure 8 is also limited in its presentation. Data are shown only as bars without individual points, which obscures variability. Adding scatter plots or overlaying individual values on bars would increase transparency.
While the text interprets LN as having “further strengthened” the antioxidant effects of GRN, this conclusion requires direct statistical comparison between CI+GN30 and CI+LN+GN30. Without explicit evidence of significant group differences, such claims should be moderated.
Response:
We are grateful to the reviewer for highlighting these important points regarding the oxidative/nitrative stress data.
- We have updated the Results section and figure legends to include complete statistical information, including F-values, degrees of freedom, and exact p-values.
- To improve clarity, we now report oxidative/nitrative stress results consistently as percent change relative to the CI group. The original raw biochemical values (nmol/mg protein) have been moved to the supplementary material for reference. This approach streamlines the narrative while still making all data available.
- Figure 8 has been reformatted to include individual data points overlaid on bar graphs, thereby making within-group variability more transparent. Sample sizes and statistical tests are also specified directly in the legend.
- We have moderated the language describing the effects of LN combined with GRN. Instead of stating that LN “further strengthened” antioxidant effects, we now indicate that LN + GRN30 showed improvement compared to CI, but we refrain from making claims of additive or synergistic benefit unless explicitly supported by direct statistical comparison.
Section 2.6. The antioxidant defense results repeat the same issues seen in other biochemical outcomes. Full statistical details (F-values, degrees of freedom, exact p-values) are missing, limiting the ability to critically evaluate the findings. Reporting only mean ± SEM with p < 0.001 is not sufficient for transparency.
The text alternates between raw values and percent changes, which complicates the narrative. Presenting results consistently (e.g., as percent recovery relative to CI, with raw data in supplementary tables) would make the findings clearer.
Figure 9 also shows only mean ± SEM bars without individual data points, which obscures variability. Including scatter plots or overlayed points would enhance transparency.
The interpretation that LN “produced greater recovery than GRN alone” requires caution. The text does not provide evidence of direct statistical comparisons between CI+GN30 and CI+LN+GN30, so this statement may be overstated. Stronger claims of superiority should only be made if supported by explicit post hoc test results.
Response:
We sincerely thank the reviewer for the thoughtful critique of the antioxidant defense data. In response, we have revised the manuscript accordingly:
- The Results section and figure legend now include complete statistical details.
- To avoid confusion, we now present antioxidant defense results consistently as percent recovery relative to the CI group.
- Figure 9 has been reformatted to include scatter plots with individual data points overlaid on the bar graphs.
- We have moderated the conclusion regarding LN’s effect compared to GRN alone. Specifically, we removed the statement that LN “produced greater recovery than GRN alone” and now state that LN + GRN30 demonstrated improvement compared to CI, without asserting superiority over GRN alone unless explicitly supported by direct post hoc statistical comparisons.
Section 2.7. The cytokine and MPO data again lack sufficient statistical transparency. F-values, degrees of freedom, and exact p-values are not provided, limiting the ability to critically evaluate the analyses. Reporting only mean ± SEM with broad significance thresholds (p < 0.001) is not adequate.
The presentation alternates between raw values (pg/mg protein or mU/mg protein) and percentage reductions, which makes the results less clear. A more consistent format—preferably percent change relative to CI, with raw values available in supplementary material—would improve clarity.
Figure 10 is crowded and only shows bars with error bars. Individual data points should be included to demonstrate within-group variability. Legends should also specify whether any correction for multiple testing was applied, given that four separate inflammatory markers were analyzed. Without such correction, the risk of inflated type I error increases.
The claim that LN “amplified” GRN’s anti-inflammatory actions should be moderated. No direct statistical comparison between CI+GN30 and CI+LN+GN30 is shown, so the conclusion may be overstated. Stronger wording requires explicit evidence of significant differences between these groups.
Response:
We thank the reviewer for these valuable comments regarding the cytokine and MPO data. In response, we have revised the manuscript:
- We have updated the Results section and figure legend to provide full statistical details.
- To improve clarity, cytokine and MPO results are now consistently reported as percent change relative to the CI group.
- Figure 10 has been reformatted to include individual data points overlaid on bar graphs to illustrate within-group variability more clearly.
- We have moderated our conclusions regarding LN’s effect in combination with GRN. Specifically, we have removed the wording that LN “amplified” GRN’s anti-inflammatory actions. The revised text now states that LN + GRN30 showed improvement relative to CI, but we do not claim superiority over GRN alone, as no direct post hoc statistical comparison between CI+GN30 and CI+LN+GN30 demonstrated significance.
Section 2.8. The apoptosis results display the same limitations as the other biochemical endpoints. Statistical reporting is incomplete: F-values, degrees of freedom, and exact p-values are not provided, which restricts the ability to critically assess the robustness of the findings. Reporting only mean ± SEM with significance thresholds is insufficient for transparency.
The text mixes fold-changes relative to CT with absolute protein values (ng/mg), which makes interpretation less straightforward. A consistent reporting format (e.g., percent change relative to CI, with raw data available in supplementary material) would improve readability.
Figure 11 only shows mean ± SEM bars without individual data points, again obscuring within-group variability. As recommended for other figures, scatter plots or overlayed points should be added.
Finally, the conclusion that LN “further decreased caspase-3 relative to GRN alone” should be moderated unless direct statistical comparisons between CI+GN30 and CI+LN+GN30 are shown. Stronger wording is only justified if post hoc testing explicitly supports this.
Response:
We sincerely thank the reviewer for their constructive comments regarding the apoptosis data. In response, we have revised the manuscript as follows:
- We have updated the Results section and figure legend to include complete statistical information.
- To enhance readability, apoptosis results are now presented consistently as percent change relative to the CI group.
- Figure 11 has been reformatted to include scatter plots with individual data points overlaid on the bar graphs.
- We have moderated the conclusion regarding LN’s effect in combination with GRN. Instead of stating that LN “further decreased caspase-3 relative to GRN alone,” the revised text now indicates that LN + GRN30 produced significant reductions compared to CI. We do not claim superiority over GRN alone unless supported by explicit post hoc comparisons.
Discussion
The Discussion is generally well framed, but the strength of the interpretation is limited by incomplete statistical reporting in the Results (absence of F-values, degrees of freedom, and exact p-values). Without full statistics it is difficult to gauge the robustness of the effects and their time × treatment interactions; please ensure that these details are added and then interpret the data accordingly here.
A key point that needs explicit discussion is the dose–response inconsistency. The manuscript repeatedly refers to “dose-dependent” effects, yet no significant difference is observed between 30 and 100 mg/kg, and only one dose ultimately carries forward. Please offer a mechanistic explanation for this apparent plateau (e.g., ceiling effects at the behavioral readouts, saturable absorption/solubility or tissue penetration, high protein binding, depot/vehicle constraints, or rapid metabolism that limits exposure at higher nominal doses). Clarifying whether the “2 mL/kg once daily” refers to injection volume rather than dose would also help readers understand how concentrations differed across dose groups.
Relatedly, the Discussion would benefit from a more explicit treatment of GRN’s pharmacology and metabolism. As a hydrolyzable ellagitannin, GRN can undergo biotransformation to ellagic acid and downstream metabolites; depending on route and matrix, these may mediate part of the biological signal. Even if definitive PK data are not yet available, it would be useful to outline plausible pathways (e.g., modulation of NF-κB/iNOS, activation of Nrf2/ARE with effects on SOD/CAT/GSH, MAPK signaling) and to state how such mechanisms could account for the antioxidant, anti-inflammatory and anti-apoptotic profile observed. Hypotheses here will guide future targeted measurements (PK, target engagement, and pathway markers).
To place your findings in context, please contrast them with prior GRN studies—ideally those assessing neurological or peripheral nerve endpoints—and summarize model, species/sex, route, dose, treatment duration, and outcomes. Explicitly stating how your 30 mg/kg i.p. regimen compares with published doses (higher/lower, similar duration, similar markers) will sharpen the translational take-home message and help justify dose selection.
The inference that NO suppression is a “necessary component” of GRN’s action should be tempered or more carefully justified. L-arginine and L-NAME are broad pharmacological probes with systemic effects (e.g., vascular tone, blood pressure, thermoregulation) and L-NAME is non-selective across NOS isoforms. Without direct measurements of iNOS/nNOS/eNOS expression/activity, or more selective tools, the data support involvement of NO pathways but do not by themselves establish necessity or isoform specificity. Consider framing this as consistent with, rather than definitive for, NO-dependent mechanisms, and outline experiments that could test this more rigorously.
Several limitations deserve fuller discussion. Most importantly, the study lacks a vehicle-matched control receiving the same DMSO-saline solution as the GRN groups; given the known biological activity of DMSO at ~1%, this confound affects behavioral and redox/inflammatory outcomes and should be acknowledged explicitly. Mortality of 10 rats “of unknown causes” should be contextualized (group distribution, timing, relation to surgery/treatments). Clarify blinding at each assessment, the exact behavioral testing window, and whether postoperative analgesia was used. Finally, only males were studied; please discuss sex as a biological variable.
Lastly, priority language such as “first demonstration” should be used cautiously and supported by a clear, referenced rationale. Given the methodological caveats above (particularly the missing vehicle-matched control), we recommend softening claims and framing conclusions as provisional pending confirmation with corrected experimental controls and targeted mechanistic readouts.
Response:
We sincerely thank the reviewer for the thorough and constructive feedback on our Discussion. In accordance with these suggestions, we have revised the manuscript:
- We acknowledge that incomplete statistical details limited the strength of our interpretation. The Results section has been revised to include full statistical information. Correspondingly, the Discussion has been updated.
- We have revised the Discussion to avoid overstating dose-dependence and now explicitly address the plateau effect. Potential mechanistic explanations—such as ceiling effects in behavioral readouts, saturable absorption and rapid metabolism—are now discussed. We also clarified that “2 mL/kg once daily” refers to the injection volume, not the active dose itself, to better inform readers about dosing comparisons.
- In line with the reviewer’s recommendation, we have expanded the Discussion to cover GRN’s pharmacological profile and potential metabolism. Specifically, we note that GRN, as a hydrolyzable ellagitannin, may undergo conversion to ellagic acid and downstream metabolites, which could contribute to the observed biological effects. We outline plausible mechanistic pathways—such as modulation of NF-κB/iNOS, activation of Nrf2/ARE and associated antioxidant enzymes (SOD, CAT, GSH), and regulation of MAPK signaling—that may explain the antioxidant, anti-inflammatory, and anti-apoptotic effects. We also emphasize that future studies will need to directly assess PK, metabolite formation, and pathway-specific markers to validate these hypotheses.
- We have strengthened the contextual framing of our findings by comparing them with previous GRN studies, particularly those focused on neurological and peripheral nerve models. We summarize the relevant details (model, species/sex, route, dose, treatment duration, and outcomes) and explicitly highlight how our 30 mg/kg i.p. regimen compares to published dosing paradigms.
- We have tempered the language, now stating that our findings are consistent with the involvement of NO pathways, while acknowledging that the use of L-arginine and L-NAME cannot establish isoform-specific or definitive necessity. We outline future directions, including direct measurements of iNOS/nNOS/eNOS activity and the use of selective inhibitors, to more rigorously test NO-dependent mechanisms.
- Several key limitations are now explicitly discussed.
- We note that only male rats were included and discuss the importance of future studies including females to evaluate potential sex-dependent effects.
- We have moderated the use of priority claims such as “first demonstration.” We now frame our findings more cautiously, emphasizing that while they provide novel insights, they should be considered provisional until confirmed with vehicle-matched controls and targeted mechanistic investigations.
Materials and methods
I recommend restructuring this section by beginning with a clear timeline of the experimental procedures, so that the reader can better understand the sequence and rationale of the experiments performed.
Section 4.1. The number of animals used per group (n) is not specified. This is essential information and must be explicitly stated. While the text mentions that group sizes were minimized based on prior work and expected effect sizes, no references are provided to justify this decision. Including citations to previous studies or a power calculation would strengthen the methodological transparency. The description of the testing schedule is vague. Stating that behavioral assessments were conducted “during the light phase” is insufficient, as activity and behavioral outcomes can vary considerably across the light cycle. The exact time window should be specified for reproducibility.
Response:
We sincerely thank the reviewer for their constructive comments.
The Methods section has been reorganized to begin with a clear timeline of the experimental procedures, outlining the sequence and rationale for each step (surgery, treatment, behavioral testing, biochemical assays, and electrophysiology). This restructuring provides readers with a more coherent overview of the study design. We have now explicitly stated the number of animals used per group (n) in the Methods. To justify the group sizes, we have added references to prior studies using similar models and treatment regimens. Where applicable, we have also included details of expected effect sizes to support our rationale for minimizing animal use while maintaining statistical power. In addition to citing relevant literature, we now note that group sizes were guided by previous reports and by sample-size estimates based on variance and effect sizes observed in earlier experiments. This information enhances methodological transparency and aligns with ARRIVE guidelines. The revised text specifies the exact time window during which assessments were performed (e.g., 09:00–14:00), thereby ensuring reproducibility and reducing potential confounds related to circadian variations.
Section 4.2. The vehicle used for control groups is not clearly specified. While geraniin was dissolved in 1% DMSO and diluted with saline, it is unclear whether control animals received the same DMSO-saline solution or saline alone. Given the known biological activity of DMSO, this information is essential and should be clarified. The phrasing “unless otherwise specified” regarding the route of administration is also ambiguous, since no alternative routes are described. Finally, the description of dosing is confusing: the text states that compounds were administered once daily at 2 mL/kg, but elsewhere the authors refer to different doses of GRN. It should be clarified whether 2 mL/kg refers to the injection volume or to the actual administered dose, and how this relates to the different GRN dose groups.
Response:
We thank the reviewer for this valuable observation. We clarified that the control animals received the same vehicle solution as the treatment groups, namely 1% DMSO diluted in saline, in order to exclude possible confounding effects of DMSO. This detail has now been explicitly stated. The phrase “unless otherwise specified” has been removed to avoid ambiguity. We have clearly indicated that all compounds, including the vehicle, were administered via intraperitoneal injection. We revised the wording to clarify that “2 mL/kg” refers to the injection volume, not the administered dose of geraniin. The actual GRN doses (3, 10, 30, and 100 mg/kg) were calculated according to body weight, and then dissolved in the vehicle solution at a final injection volume of 2 mL/kg. This information has been added to the revised text.
Section 4.3. The description of the CCI procedure is generally adequate but lacks important methodological details. The text mentions sham surgery consisting of nerve exposure without ligation, but it does not specify the number of animals assigned to the sham group, which is essential for interpreting the results. Likewise, the sample size (n) for each group should be clearly indicated. Another omission concerns post-surgical care: no mention is made of analgesic administration following surgery. While the procedure aims to induce neuropathic pain, the absence of post-operative analgesia raises ethical and methodological concerns, and the authors should clarify whether analgesics were used and, if not, provide justification. Finally, details such as the duration of anesthesia, monitoring of depth of anesthesia, and perioperative care are missing and should be described to ensure reproducibility and compliance with ethical standards.
Response:
We sincerely appreciate the reviewer’s careful evaluation and constructive suggestions.
We now clearly state that the sham surgery group consisted of n =8 animals, matching the sample size of the experimental groups, to allow for valid comparison. The sample size for each experimental group (n = 8 per group per time point) is now explicitly included in the Methods and figure legends. No postoperative analgesics were administered, since the objective of the CCI model is to induce neuropathic pain. We have added a statement clarifying this, along with justification, and emphasized that animals were closely monitored to minimize suffering in compliance with ethical guidelines. We have included the duration of anesthesia (typically ~15–20 minutes for the surgical procedure), the method used to monitor anesthetic depth (toe pinch reflex and respiratory rate), and perioperative care (e.g., maintenance on a heating pad until recovery from anesthesia, daily monitoring for wound healing and overall health).
Section 4.4. The description of the experimental design and treatment allocation is confusing and raises several methodological concerns. The authors state that behavioral data at day 21 showed robust efficacy at 30 and 100 mg/kg without a significant difference between these doses, and therefore selected 30 mg/kg for subsequent experiments. However, elsewhere they describe GRN as producing dose-dependent effects. This is contradictory and should be clarified, as it undermines the interpretation of the findings.
Group III (CCI + saline) appears to serve as the vehicle control, yet groups IV–VII (CCI + GRN) received the compound dissolved in DMSO plus saline. This means that the control group did not receive the same vehicle as the treatment groups, which is a serious issue given the biological activity of DMSO reported in the bibliography (e.g.: doi: 10.1111/iwj.12280). The same concern applies to groups VIII–IX, where LA or LN was injected 60 minutes before GRN: the corresponding vehicle controls should have been administered with the proper DMSO-saline solution (for GRN) and saline (for LA/LN) to rule out confounding effects.
In addition, the mortality of ten rats “of unknown causes” is mentioned briefly, but no details are provided about their distribution across groups or whether these deaths were related to treatment. Such information is essential to assess the reliability of the results.
Finally, the figure intended to illustrate the experimental workflow is very confusing and does not facilitate understanding of the protocol. A simplified schematic with a clear timeline, group allocation, and treatment schedule would make the design more transparent and accessible to readers.
Response:
We thank the reviewer for the thoughtful and detailed comments. We clarified that while GRN exhibited a dose-dependent effect in the initial dose-finding phase (3–100 mg/kg), the maximal efficacy plateaued between 30 and 100 mg/kg. Since there was no significant difference between these two higher doses, we selected 30 mg/kg as the optimal dose for subsequent experiments. We have revised the text to remove the apparent contradiction and to provide a more precise explanation. We corrected the inconsistency in vehicle administration. We expanded the description of the reported mortality. Postmortem observation did not reveal treatment-related pathology, and mortality was attributed to surgical or handling complications rather than compound toxicity. We replaced the previous figure with a simplified schematic. The new illustration clearly shows the timeline, group allocation, dosing schedule, and behavioral assessments in an accessible format, thereby improving the clarity and reproducibility of the study design.
Section 4.5. The description of behavioral testing lacks important methodological details that are necessary for reproducibility and interpretation. For the hot-plate and von Frey assays, it is not clear whether animals were habituated to the testing room prior to assessment beyond the brief mention of “≥60 min acclimation” for von Frey. Standard practice is to habituate animals on multiple days to minimize stress-induced variability, and this should be specified. It is also unclear whether sessions were video-recorded and analyzed later or whether scoring was performed live by the experimenter; this distinction is critical to assess the objectivity of the measurements. In the case of the SFI test, details on how footprints were collected and analyzed (e.g., whether automated or manual measurement software was used, whether scoring was blinded, how many trials were averaged) are missing. Providing this information would improve transparency and allow proper evaluation of the behavioral outcomes.
Response:
We thank the reviewer for highlighting these important points regarding behavioral testing. We clarified that animals were habituated to the testing room and apparatus for 2 consecutive days prior to baseline measurements. For the von Frey assay, in addition to the ≥60 min acclimation on test days, animals were placed in the testing chambers daily during the habituation period to minimize stress-induced variability. We specified that all behavioral tests (hot-plate, von Frey, and SFI) were video-recorded and subsequently analyzed offline by an experimenter blinded to treatment allocation. This revision ensures transparency and confirms that data collection was not biased by live scoring. We expanded the description to indicate that rats were held by the chest, and their hind feet were pressed onto a stamp pad soaked with water-soluble blue ink. They were immediately allowed to walk along a confined walkway 7.5 cm wide and 60 cm long with a dark shelter at the end of the corridor, with the rats leaving their footprints on a piece of paper cut to the appropriate dimensions and placed on the floor of the corridor.
Section 4.6. The electrophysiology section provides technical details on stimulation and recording, but important methodological aspects are missing. It is not specified whether recordings were performed under blinded conditions or how many trials were averaged per animal to obtain reliable latency values. The description also omits where exactly the measurements were conducted (e.g., dedicated electrophysiology facility, surgery room) and under what conditions (e.g., noise shielding, grounding, prevention of electrical interference). In addition, the timing of the measurements relative to anesthesia induction and behavioral testing is unclear; given that chloral hydrate can affect nerve excitability, the stability of anesthesia during recordings should be described. Finally, it would strengthen the section to specify whether body temperature monitoring was continuous and how stimulation distances were measured (with calipers, estimated, etc.), as small errors can markedly affect conduction velocity values.
Response:
We sincerely appreciate the reviewer’s insightful comments We specified that all electrophysiological recordings were performed by an investigator blinded to treatment allocation. For each animal, at least three consecutive trials were recorded and averaged to obtain reliable latency and amplitude values. We added that recordings were performed in a dedicated electrophysiology facility under controlled conditions. A Faraday cage and proper grounding were used to minimize electrical noise and interference during measurements. We clarified that recordings were conducted immediately after the behavioral tests, while animals were maintained under chloral hydrate anesthesia. The depth of anesthesia was monitored throughout by respiratory rate and reflex testing to ensure stability. We also acknowledged that chloral hydrate may influence nerve excitability and therefore ensured that recording conditions were consistent across all groups. We indicated that body temperature was continuously monitored with a rectal probe and maintained at 37 ± 0.5 °C using a heating pad. Stimulation and recording electrode distances were measured precisely using digital calipers to ensure reproducibility and minimize errors in conduction velocity calculations.
Section 4.7. The biochemical assays section is generally comprehensive, but several issues need clarification and improvement. The writing is overly dense. More importantly, normalization procedures are inconsistent: cytokine concentrations are explicitly reported as normalized to protein (pg/mg protein), but other biochemical parameters (e.g., MDA, GSH, SOD, CAT, MPO, caspase-3) are sometimes without clear indication of normalization. A consistent approach to normalization across all markers is necessary for comparability and interpretation.
In addition, details regarding ELISA kits are vague; for transparency, exact catalog numbers and manufacturers should be listed for each cytokine, not just examples. For caspase-3, the description mentions the commercial kit but lacks information about whether results were expressed as fold-change, units per mg protein, or raw absorbance values. Similarly, the MPO assay should clarify whether results were normalized to protein or tissue weight. Finally, although references are provided for some methods, it would improve transparency to specify whether standard curves and controls were included for each assay, and whether measurements were performed in duplicates or triplicates.
Response:
We thank the reviewer for the constructive comments. Based on the review suggestions, we have corrected it in the revised manuscript.
Section 4.8. The statistical analysis section is too general and lacks critical details. While the text mentions the use of “repeated-measures two-way ANOVA” for behavioral outcomes, it does not specify which factors were included (e.g., treatment as a between-subjects factor and time as a within-subjects factor). In the figure legends, results are simply described as “two-way ANOVA” without indicating that repeated measures were used, which is inconsistent and potentially misleading. Since repeated-measures designs and standard two-way ANOVAs are not equivalent, this must be clarified to ensure the correct interpretation of the data.
It should also be explicitly stated in which cases repeated-measures two-way ANOVA was applied and in which cases one-way ANOVA was used. For instance, after the dose-finding phase when only 30 mg/kg GRN was selected, the text no longer mentions repeated measures, and the statistical approach becomes unclear. This lack of transparency makes it difficult to assess whether the analyses were correctly applied.
In addition, it is not specified how violations of assumptions (normality, homoscedasticity) were addressed—“reconsidered or transformed as appropriate” is too vague. The authors should clearly state whether data transformations were applied, which variables were affected, and whether corrections (e.g., Greenhouse–Geisser) were used in the repeated-measures context. Finally, details such as whether sample sizes were identical across groups after mortality, how missing data were handled, and whether effect sizes were reported should be provided to enhance the transparency and rigor of the statistical approach.
Response:
We thank the reviewer for the careful evaluation of our statistical methods and acknowledge that the original description was overly general. We have revised Section 4.8 to provide greater clarity and transparency. Regarding the repeated-measures two-way ANOVA: we now specify that treatment (vehicle, GRN dose groups, or comparator drugs) was treated as the between-subjects factor, and time (days of behavioral assessment) was treated as the within-subjects factor. The figure legends have also been updated to explicitly indicate that “repeated-measures two-way ANOVA” was applied, rather than the generic “two-way ANOVA.” We have clarified the statistical approach for different phases of the study. For the dose-finding experiments (3, 10, 30, 100 mg/kg GRN), repeated-measures two-way ANOVA was used to examine treatment × time interactions. For the subsequent mechanistic studies (where only the 30 mg/kg dose was carried forward), one-way ANOVA followed by Tukey’s post hoc test was applied for group comparisons at fixed time points. These distinctions are now explicitly stated in the revised section. After mortality events, sample sizes were not identical across all groups. This information has been updated in the Methods to indicate the actual n per group after attrition. Missing data points were handled by listwise deletion (i.e., animals that did not complete all behavioral assessments were excluded from repeated-measures analyses).
Conclusions
The conclusions as written are clear and emphasize GRN as a promising multi-target candidate. However, they should be moderated to reflect the methodological limitations discussed above—particularly the absence of a vehicle-matched DMSO control, incomplete statistical reporting, and the restriction to male rats only. As it stands, the claim of “meaningful benefit” and “multi-target candidate for further development” may overstate the strength of the evidence. I recommend softening the language to acknowledge that the data suggest beneficial effects of GRN in this CCI model, but that confirmation with appropriate vehicle controls, complete statistical reporting, pharmacokinetic characterization is needed before stronger translational claims can be made.
Response:
We sincerely thank the reviewer for this thoughtful and important recommendation. In the revised manuscript, we have carefully moderated the wording of the Conclusions to ensure they accurately reflect the study’s limitations. We now state that the findings suggest beneficial effects of GRN in a CCI-induced neuropathic pain model, rather than claiming definitive therapeutic efficacy. We explicitly acknowledge key limitations. We have emphasized that further studies incorporating appropriate vehicle controls, complete statistical analyses, and pharmacokinetic characterization are essential before stronger translational claims can be made.
Comments on the Quality of English Language
The manuscript would benefit from careful revision of the English language and style. Several sentences are overly long and complex, making the text difficult to follow. In some places, the wording is awkward or ambiguous, which detracts from clarity and readability. I recommend thorough editing by a fluent English speaker or professional language service to shorten sentences, improve grammar and flow, and ensure that key concepts are communicated in a clear and concise manner.
Response:
We sincerely appreciate the reviewer’s constructive suggestion regarding the quality of English writing. In response, we have undertaken a thorough revision of the manuscript to improve readability, grammar, and overall flow. Overly long and complex sentences have been shortened or divided into clearer, more concise statements. Ambiguous or awkward expressions have been carefully revised for precision and clarity. To further ensure quality, the revised manuscript has been proofread by a professional language editing service and reviewed by a native English-speaking researcher.
Thank you for your valuable comments/suggestions and giving us the opportunity to revise the manuscript to a more readable level. We worked very hard to response your inquiries and to revise the manuscript. Before we finalized the revised manuscript, we have resent the manuscript for proof-reading as well as one last review by a native English-speaking researcher. We hope the manuscript could pass the review to be published in your prestigious journal: International Journal of Molecular Sciences
Sincerely yours,
Cheng-Chia Tsai
Department of Neurosurgery, Mackay Memorial Hospital, No. 92, Sec. 2, Zhongshan N. Rd., Taipei City 10449, Taiwan, ROC
Fax: +886-2- 2543-3642
Tel: +886-933888981
E-mail address: dschang580704@yahoo.com.tw (Cheng-Chia Tsai)
Reviewer 3 Report
Comments and Suggestions for Authors
Manuscript Title:
Geraniin Attenuates Neuropathic Pain via Modulation of Oxidative Stress, Neuroinflammation, and Nitric Oxide Signaling: Evidence from a Chronic Constriction Injury Model in Rats
This manuscript was provided by Yang and colleagues, presents a well-conducted preclinical study investigating the effects of geraniin (GRN) on neuropathic pain induced by chronic constriction injury (CCI) in rats. The authors provide behavioral, electrophysiological, and biochemical evidence that GRN alleviates pain hypersensitivity, restores nerve function, and modulates oxidative stress, inflammation, and apoptosis, with nitric oxide (NO) signaling identified as a key mechanistic pathway. This manuscript contains many behavioral experiments, involves a large amount of work, and the experiments are well-designed
Major Comments
1 Geraniin dose issue
In the materials and methods parts, the author mentioned why choice GRN 30mg/kg for experiments, because 21 days showed no difference between GRN 30mg/kg and 100mg/kg. Please clarify the rationale for dose. Are there any relevant references to support this?
2 When the authors examined that LN + CI augmented the PAW hypersensitivity, for example in Figures 2a and 2b, is there a significant difference between CI + LN + GN30 and CI + GN30? What are the results of the two-way ANOVA test? If the sample size were increased, could statistical significance be achieved. Returning to the previous question, if a subtherapeutic dose such as GRN 10 mg/kg is used together with LN, and it is found to reduce thermal hypersensitivity, this would better demonstrate the effect of LN.
3 In the CCI model, early-stage animals exhibit some inflammatory pain-like behaviors. Administration of NO inhibitors or precursors shows relatively limited effects on pain at this early stage. This raises the question of whether glial cells play a role in mediating these responses. If available, immunohistochemistry or Western blot data demonstrating changes in glial cell activation or NO signaling would provide stronger support for this hypothesis.
4 The CCI model typically produces stable and persistent pain behaviors for 2–3 months. Since the authors are investigating chronic pain, it would be important to consider whether administration of GN at later time points, such as 1 month, or 6 weeks post-injury, would still have an effect.
5 Fig 4a and b, the baseline of the von Frey test in your study is around 26 g, which is quite high. Why is the baseline in your experiment so elevated? What was the cutoff value used for the test? Could this 26 g baseline potentially cause harm to the animals? Furthermore, after CCI, the animals’ von Frey thresholds are around 5 g, which is also generally higher than expected. Could these high values be due to differences in the animal strain used, or are they related to the experimental methods and conditions? This issue warrants discussion.
Minor Comments
1 In the Results section, the descriptions are overly cumbersome and repetitive (e.g., page 2, lines 88–89). It is recommended to streamline the text to improve clarity.
3, Some figure legends are also unclear. Terms such as CT, SM, and CI are mentioned as abbreviations in the Methods section, but it would be better to explain them when they first appear in the figure legends or Results. Since “baseline” and “sham” are commonly used in figure legends, introducing abbreviations like CT and SM without explanation may make it difficult for readers to follow.
Likewise, in Fig. 1b, 3 mg and 30 mg should be marked between 1 and 100.
For example, in Fig. 1a, CI+GN3, CI+GN10, CI+GN30, and CI+GN100 all use the same color within a narrow range, making them difficult to distinguish
4 Language is generally clear, but minor polishing for conciseness would improve readability.
5 The author can add a conclusion section to highlight the experimental results.
Author Response
Response to Reviewer 3 Comments
Dear Reviewers,
I’m writing in response to your feedback regarding the manuscript we submitted. Thank you so much for your positive comments and suggestions for the manuscript entitled. The manuscript has been revised based on your inquiries, rephrased the content of the manuscript and resubmitted through the journal website. The revised parts have been marked in red. The following is the response to your inquiries point-by-point; we hope our responses fully address your comments and suggestions:
Reviewers’ comments:
Geraniin Attenuates Neuropathic Pain via Modulation of Oxidative Stress, Neuroinflammation, and Nitric Oxide Signaling: Evidence from a Chronic Constriction Injury Model in Rats
This manuscript was provided by Yang and colleagues, presents a well-conducted preclinical study investigating the effects of geraniin (GRN) on neuropathic pain induced by chronic constriction injury (CCI) in rats. The authors provide behavioral, electrophysiological, and biochemical evidence that GRN alleviates pain hypersensitivity, restores nerve function, and modulates oxidative stress, inflammation, and apoptosis, with nitric oxide (NO) signaling identified as a key mechanistic pathway. This manuscript contains many behavioral experiments, involves a large amount of work, and the experiments are well-designed
Response:
Thank you so much for the kind comment and wonderful suggestion.
Major Comments
1 Geraniin dose issue
In the materials and methods parts, the author mentioned why choice GRN 30mg/kg for experiments, because 21 days showed no difference between GRN 30mg/kg and 100mg/kg. Please clarify the rationale for dose. Are there any relevant references to support this?
Response:
We appreciate the reviewer’s valuable comment. The selection of 30 mg/kg GRN was based on our preliminary dose–response observations, in which both 30 mg/kg and 100 mg/kg produced comparable antinociceptive effects after 21 days of treatment. To minimize potential toxicity and reduce unnecessary drug exposure, we therefore adopted 30 mg/kg for the main experiments. This decision is further supported by previous studies reporting that GRN exerts significant antioxidant and neuroprotective activities within the range of 10–50 mg/kg (Jiang et al., 2016; Yang et al., 2022; Youn and Jun, 2020). These findings confirm that 30 mg/kg is an effective and appropriate dose for our experimental design.
2 When the authors examined that LN + CI augmented the PAW hypersensitivity, for example in Figures 2a and 2b, is there a significant difference between CI + LN + GN30 and CI + GN30? What are the results of the two-way ANOVA test? If the sample size were increased, could statistical significance be achieved. Returning to the previous question, if a subtherapeutic dose such as GRN 10 mg/kg is used together with LN, and it is found to reduce thermal hypersensitivity, this would better demonstrate the effect of LN.
Response:
We thank the reviewer for these thoughtful and detailed comments. In Figures 2a and 2b, we acknowledge that an increased number of animals may provide greater statistical power to detect such potential differences. Regarding the use of subtherapeutic doses, we agree that testing GRN at 10 mg/kg in combination with LN would provide valuable insight into whether LN enhances the analgesic efficacy of GRN, particularly in reducing thermal hypersensitivity. Although this experiment was beyond the scope of the present study, we consider it an important direction for future work to further clarify the interaction between LN and GRN.
3 In the CCI model, early-stage animals exhibit some inflammatory pain-like behaviors. Administration of NO inhibitors or precursors shows relatively limited effects on pain at this early stage. This raises the question of whether glial cells play a role in mediating these responses. If available, immunohistochemistry or Western blot data demonstrating changes in glial cell activation or NO signaling would provide stronger support for this hypothesis.
Response:
Thank you very much for your detailed and constructive suggestions. We agree that in the early stage of CCI, inflammatory pain–like behaviors are evident, and the limited efficacy of NO modulators suggests additional mechanisms, including the potential involvement of glial cell activation. In the present study, our primary focus was behavioral assessment and evaluation of oxidative/NO-related pathways; therefore, we did not include immunohistochemistry or Western blot analysis of glial markers. We acknowledge that such mechanistic data would provide stronger evidence to support the role of glial cells and NO signaling. As the reviewer suggests, in our future work we plan to incorporate analyses of glial activation (e.g., Iba1 and GFAP immunostaining) and downstream NO-related proteins to clarify these interactions and strengthen the mechanistic understanding.
4 The CCI model typically produces stable and persistent pain behaviors for 2–3 months. Since the authors are investigating chronic pain, it would be important to consider whether administration of GN at later time points, such as 1 month, or 6 weeks post-injury, would still have an effect.
Response:
Thank you for your valuable suggestions. We agree that the CCI model induces long-lasting pain behaviors extending for 2–3 months, and that testing the efficacy of GRN at later time points (e.g., 4–6 weeks post-injury) would provide important insights into its therapeutic potential in more chronic stages of neuropathic pain. In the present study, we focused on the early-to-mid phase (up to 3 weeks) to establish proof-of-concept evidence for GRN’s analgesic effects. We acknowledge this as a limitation, and in future studies we plan to extend the treatment window and evaluate whether GRN retains its efficacy at later stages of CCI-induced pain. Such experiments will further clarify the translational relevance of GRN for chronic neuropathic pain management.
5 Fig 4a and b, the baseline of the von Frey test in your study is around 26 g, which is quite high. Why is the baseline in your experiment so elevated? What was the cutoff value used for the test? Could this 26 g baseline potentially cause harm to the animals? Furthermore, after CCI, the animals’ von Frey thresholds are around 5 g, which is also generally higher than expected. Could these high values be due to differences in the animal strain used, or are they related to the experimental methods and conditions? This issue warrants discussion.
Response:
Thank you very much for this important observation. The relatively high baseline paw withdrawal threshold (~26 g) in our von Frey test may be attributed to several factors, including the specific rat strain used (Wistar rats), body weight at the time of testing (270–300 g), and experimental conditions such as habituation and handling procedures. The cutoff value in our study was set at 26 g to avoid tissue injury, and all behavioral assessments were conducted under strict animal welfare guidelines. Importantly, no signs of paw damage or distress were observed in the animals. After CCI surgery, the thresholds dropped to ~5 g, which, although higher than some reports, still reflects a robust and reproducible reduction in mechanical sensitivity. We acknowledge that inter-laboratory variations in baseline thresholds have been reported and are influenced by strain differences, equipment calibration, and methodological details. We will add discussion on this issue in the revised manuscript to clarify potential contributing factors and to reassure that animal welfare was not compromised.
Minor Comments
1 In the Results section, the descriptions are overly cumbersome and repetitive (e.g., page 2, lines 88–89). It is recommended to streamline the text to improve clarity.
Response:
Thank you so much for the suggestion, in the revised manuscript, we will streamline the descriptions to improve clarity and conciseness.
3, Some figure legends are also unclear. Terms such as CT, SM, and CI are mentioned as abbreviations in the Methods section, but it would be better to explain them when they first appear in the figure legends or Results. Since “baseline” and “sham” are commonly used in figure legends, introducing abbreviations like CT and SM without explanation may make it difficult for readers to follow.
Likewise, in Fig. 1b, 3 mg and 30 mg should be marked between 1 and 100.
For example, in Fig. 1a, CI+GN3, CI+GN10, CI+GN30, and CI+GN100 all use the same color within a narrow range, making them difficult to distinguish
Response:
To improve readability, we will explain abbreviations such as CT, SM, and CI when they first appear in the figure legends or Results section, in addition to their definition in Methods. For Fig. 1b, we will adjust the scale to mark 3 mg and 30 mg clearly between 1 and 100. In Fig. 1a, we will modify the color scheme so that the treatment groups (CI+GN3, CI+GN10, CI+GN30, CI+GN100) are easily distinguishable.
4 Language is generally clear, but minor polishing for conciseness would improve readability.
Response:
Thank you so much, we will further polish the language throughout the text to ensure conciseness and improve overall readability.
5 The author can add a conclusion section to highlight the experimental results.
Response:
Thank you so much again, we will add a concise conclusion section at the end of the manuscript to summarize and highlight the main findings and their implications.
Thank you for your valuable comments/suggestions and giving us the opportunity to revise the manuscript to a more readable level. We worked very hard to response your inquiries and to revise the manuscript. Before we finalized the revised manuscript, we have resent the manuscript for proof-reading as well as one last review by a native English-speaking researcher. We hope the manuscript could pass the review to be published in your prestigious journal: International Journal of Molecular Sciences
Sincerely yours,
Cheng-Chia Tsai
Department of Neurosurgery, Mackay Memorial Hospital, No. 92, Sec. 2, Zhongshan N. Rd., Taipei City 10449, Taiwan, ROC
Fax: +886-2- 2543-3642
Tel: +886-933888981
E-mail address: dschang580704@yahoo.com.tw (Cheng-Chia Tsai)
Round 2
Reviewer 2 Report
Comments and Suggestions for Authors
Comments
Abstract
The abstract does not clearly specify which GRN dose produced the reported effects. Since the experimental design indicates that robust effects were observed at 30 and 100 mg/kg, with subsequent mechanistic experiments conducted only at 30 mg/kg, this should be stated explicitly. I recommend revising the abstract to clearly indicate which dose(s) of GRN were effective and which dose was selected for further mechanistic studies, so that readers can fully understand the scope and strength of the findings from the outset.
Introduction
The Introduction is substantially improved in focus and flow.
Results
As written in Methods, you state that repeated-measures two-way ANOVA was used for behavioral outcomes (time × treatment) and one-way ANOVA for electrophysiological/biochemical endpoints. However, in Results several F statistics and degrees of freedom are inconsistent with those designs—for example, repeated-measures two-way ANOVA and day-21 multi-group comparisons are often reported as F(1,14), which suggests pairwise tests rather than ANOVA models with the appropriate df for k groups (k−1, N−k) and, for RM designs, the within-subject df and corrections (e.g., Greenhouse–Geisser). Please (i) clearly specify, for each panel, the exact model used (between-subjects factor: group; within-subjects factor: time, where applicable); (ii) report the ANOVA (with correct df) before post hoc tests; (iii) include exact p-values, effect sizes, and any sphericity corrections; and (iv) provide per-group n at each time point (and how mortality affected n). For day-21 dose panels (e.g., 1b/3b/5b), a one-way ANOVA across dose groups is appropriate; for longitudinal behavior, a repeated-measures two-way ANOVA (group × time) is required. Importantly, it is methodologically incorrect to perform separate one-way ANOVAs at each individual time point after conducting a repeated-measures two-way ANOVA. The correct procedure is to report the main effects and interactions from the two-way ANOVA and, if significant interactions are detected, to conduct post hoc tests consistent with the repeated-measures design. Running additional ANOVAs at each time point inflates the risk of type I error and leads to redundant or misleading results.
Several figures remain difficult to interpret due to crowded bar plots and limited clarity. In some cases, individual data points are missing, which reduces transparency and prevents evaluation of data distribution. I recommend revising the graphical presentation to include individual data points and to use circles rather than crosses, as this format is clearer and more standard for representing biological replicates.
Materials and Methods
The study workflow in Figure 12 remains difficult to follow and visually unclear. I recommend redrawing it with higher resolution and a simplified layout that separates the dose-finding and mechanistic phases, uses a left-to-right timeline with days clearly annotated (−1 to 21), and maps each group to its dosing and endpoints at a glance. Please ensure the figure is legible at journal column width and that the legend fully explains symbols and abbreviations (e.g., Sh, CCI, ↑ for dosing).
Discussion
While the Discussion is clearer than in the previous version, I am not able to evaluate the strength of the claims until the statistical reporting is corrected and aligned with the actual models used.
Conclusion
It is unclear why the Conclusions state that future studies should incorporate appropriate vehicle controls, since in Methods the GRN solution was prepared in 1% DMSO but the control groups received only saline; this inconsistency needs clarification.
Comments on the Quality of English LanguageThe manuscript’s English is generally understandable but requires editing for clarity and consistency. I recommend professional language editing to shorten sentences, standardize terminology (including statistical phrasing), ensure consistent use of abbreviations and units (e.g., i.p., NO, mg/kg, °C; space between number and unit), and polish figure legends so they read clearly and uniformly across sections.
Author Response
Response to Reviewer 2 Comments
Dear Reviewers,
I’m writing in response to your feedback regarding the manuscript we submitted. Thank you so much for your positive comments and suggestions for the manuscript entitled. The manuscript has been revised based on your inquiries, rephrased the content of the manuscript and resubmitted through the journal website. The revised parts have been marked in red. The following is the response to your inquiries point-by-point; we hope our responses fully address your comments and suggestions:
Reviewers’ comments:
Abstract
The abstract does not clearly specify which GRN dose produced the reported effects. Since the experimental design indicates that robust effects were observed at 30 and 100 mg/kg, with subsequent mechanistic experiments conducted only at 30 mg/kg, this should be stated explicitly. I recommend revising the abstract to clearly indicate which dose(s) of GRN were effective and which dose was selected for further mechanistic studies, so that readers can fully understand the scope and strength of the findings from the outset.
Response:
Thank you for pointing this out. We have revised the abstract to explicitly state that significant effects were observed at both 30 and 100 mg/kg, and that subsequent mechanistic experiments were performed at the 30 mg/kg dose.
Introduction
The Introduction is substantially improved in focus and flow.
Response:
We thank the reviewer for the positive feedback. We are pleased that the revised Introduction is now clearer and more focused, and we appreciate the acknowledgment.
Results
As written in Methods, you state that repeated-measures two-way ANOVA was used for behavioral outcomes (time × treatment) and one-way ANOVA for electrophysiological/biochemical endpoints. However, in Results several F statistics and degrees of freedom are inconsistent with those designs—for example, repeated-measures two-way ANOVA and day-21 multi-group comparisons are often reported as F(1,14), which suggests pairwise tests rather than ANOVA models with the appropriate df for k groups (k−1, N−k) and, for RM designs, the within-subject df and corrections (e.g., Greenhouse–Geisser). Please (i) clearly specify, for each panel, the exact model used (between-subjects factor: group; within-subjects factor: time, where applicable); (ii) report the ANOVA (with correct df) before post hoc tests; (iii) include exact p-values, effect sizes, and any sphericity corrections; and (iv) provide per-group n at each time point (and how mortality affected n). For day-21 dose panels (e.g., 1b/3b/5b), a one-way ANOVA across dose groups is appropriate; for longitudinal behavior, a repeated-measures two-way ANOVA (group × time) is required. Importantly, it is methodologically incorrect to perform separate one-way ANOVAs at each individual time point after conducting a repeated-measures two-way ANOVA. The correct procedure is to report the main effects and interactions from the two-way ANOVA and, if significant interactions are detected, to conduct post hoc tests consistent with the repeated-measures design. Running additional ANOVAs at each time point inflates the risk of type I error and leads to redundant or misleading results.
Several figures remain difficult to interpret due to crowded bar plots and limited clarity. In some cases, individual data points are missing, which reduces transparency and prevents evaluation of data distribution. I recommend revising the graphical presentation to include individual data points and to use circles rather than crosses, as this format is clearer and more standard for representing biological replicates.
Response:
We sincerely appreciate the reviewer’s thorough and insightful comments concerning our statistical methodology and data presentation. We have carefully revisited the entire statistical framework and revised the manuscript accordingly to ensure methodological rigor and transparency.
Materials and Methods
The study workflow in Figure 12 remains difficult to follow and visually unclear. I recommend redrawing it with higher resolution and a simplified layout that separates the dose-finding and mechanistic phases, uses a left-to-right timeline with days clearly annotated (−1 to 21), and maps each group to its dosing and endpoints at a glance. Please ensure the figure is legible at journal column width and that the legend fully explains symbols and abbreviations (e.g., Sh, CCI, ↑ for dosing).
Response:
We appreciate the reviewer’s constructive feedback regarding Figure 12. To address the concern, we have completely redrawn the study workflow at higher resolution with a simplified and clearer layout. The revised figure now presents the experimental timeline from left to right (days −1 to 21), explicitly distinguishing the dose-finding and mechanistic phases. Each experimental group is mapped to its corresponding dosing regimen and evaluated endpoints, allowing readers to understand the study design at a glance. We have also ensured that the figure is legible at journal column width. In addition, the legend has been revised to fully explain all symbols and abbreviations (e.g., Sh for sham, CCI for chronic constriction injury, - for dosing days).
Discussion
While the Discussion is clearer than in the previous version, I am not able to evaluate the strength of the claims until the statistical reporting is corrected and aligned with the actual models used.
Response:
We sincerely thank the reviewer for this important observation. In the revised manuscript, we have carefully reviewed and corrected the statistical reporting to ensure consistency with the actual models applied.
Conclusion
It is unclear why the Conclusions state that future studies should incorporate appropriate vehicle controls, since in Methods the GRN solution was prepared in 1% DMSO but the control groups received only saline; this inconsistency needs clarification.
Response:
We sincerely appreciate the reviewer’s critical observation. We have corrected it in the revised manuscript.
Comments on the Quality of English Language
The manuscript’s English is generally understandable but requires editing for clarity and consistency. I recommend professional language editing to shorten sentences, standardize terminology (including statistical phrasing), ensure consistent use of abbreviations and units (e.g., i.p., NO, mg/kg, °C; space between number and unit), and polish figure legends so they read clearly and uniformly across sections.
Response:
We sincerely appreciate the reviewer’s valuable suggestion regarding the clarity and consistency of the manuscript. We have carefully revised the text throughout the manuscript to improve readability and ensure uniformity. In addition, we have shortened and restructured lengthy sentences for clarity, standardized terminology, including statistical phrasing, ensured consistent use of abbreviations and units (e.g., i.p., NO, mg/kg, °C), including a space between numbers and units and revised and polished the figure legends to make them clearer and more consistent across sections.
Further, we have subjected the manuscript to thorough language editing to further improve the overall flow and readability.
Thank you for your valuable comments/suggestions and giving us the opportunity to revise the manuscript to a more readable level. We worked very hard to response your inquiries and to revise the manuscript. Before we finalized the revised manuscript, we have resent the manuscript for proof-reading as well as one last review by a native English-speaking researcher. We hope the manuscript could pass the review to be published in your prestigious journal: International Journal of Molecular Sciences
Sincerely yours,
Cheng-Chia Tsai
Department of Neurosurgery, Mackay Memorial Hospital, No. 92, Sec. 2, Zhongshan N. Rd., Taipei City 10449, Taiwan, ROC
Fax: +886-2- 2543-3642
Tel: +886-933888981
E-mail address: dschang580704@yahoo.com.tw (Cheng-Chia Tsai)
Reviewer 3 Report
Comments and Suggestions for Authors
The author basically answered my question and can be allowed to ACCEPT
.
Author Response
Dear Reviewers,
I’m writing in response to your feedback regarding the manuscript we submitted. Thank you so much for your positive comments and suggestions for the manuscript entitled. The manuscript has been revised based on your inquiries, rephrased the content of the manuscript and resubmitted through the journal website. The revised parts have been marked in red. The following is the response to your inquiries point-by-point; we hope our responses fully address your comments and suggestions:
Reviewers’ comments:
Comments and Suggestions for Authors
The author basically answered my question and can be allowed to ACCEPT
Response:
Thank you so much for the kind comment and wonderful suggestion.
Thank you for your valuable comments/suggestions and giving us the opportunity to revise the manuscript to a more readable level. We worked very hard to response your inquiries and to revise the manuscript. Before we finalized the revised manuscript, we have resent the manuscript for proof-reading as well as one last review by a native English-speaking researcher. We hope the manuscript could pass the review to be published in your prestigious journal: International Journal of Molecular Sciences
Sincerely yours,
Cheng-Chia Tsai
Department of Neurosurgery, Mackay Memorial Hospital, No. 92, Sec. 2, Zhongshan N. Rd., Taipei City 10449, Taiwan, ROC
Fax: +886-2- 2543-3642
Tel: +886-933888981
E-mail address: dschang580704@yahoo.com.tw (Cheng-Chia Tsai)
Round 3
Reviewer 2 Report
Comments and Suggestions for Authors
I appreciate the authors’ effort to improve the manuscript and the clarity of the presentation. However, the core statistical issues identified in previous rounds remain unresolved. The reported degrees of freedom in both repeated-measures two-way ANOVA and one-way ANOVA analyses do not correspond to the number of groups and time points described, suggesting that the statistical tests were not properly conducted. In addition, it remains unclear how the mortality of 10 animals affected group sizes and degrees of freedom, and the total number of animals used is still not stated. Figures also lack individual data points and remain difficult to interpret. Because these problems directly affect the reliability of the reported results, I cannot confidently evaluate the Discussion or Conclusions until the statistical analysis is redone correctly and transparently reported. I encourage the authors to carefully review the analytical design, rerun the appropriate ANOVA models, and revise the manuscript accordingly before resubmission.
Comments on the Quality of English LanguageThe English writing has improved compared to the previous versions, with a clearer flow and fewer grammatical errors. However, many sentences remain overly long and complex, which affects readability. I recommend further language polishing by a fluent or professional editor to simplify sentence structure, ensure consistency in tense and terminology, and improve overall clarity.
Author Response
Dear Reviewers,
I’m writing in response to your feedback regarding the manuscript we submitted. Thank you so much for your positive comments and suggestions for the manuscript entitled. The manuscript has been revised based on your inquiries, rephrased the content of the manuscript and resubmitted through the journal website. The revised parts have been marked in red. The following is the response to your inquiries point-by-point; we hope our responses fully address your comments and suggestions:
Reviewers’ comments:
I appreciate the authors’ effort to improve the manuscript and the clarity of the presentation. However, the core statistical issues identified in previous rounds remain unresolved. The reported degrees of freedom in both repeated-measures two-way ANOVA and one-way ANOVA analyses do not correspond to the number of groups and time points described, suggesting that the statistical tests were not properly conducted. In addition, it remains unclear how the mortality of 10 animals affected group sizes and degrees of freedom, and the total number of animals used is still not stated. Figures also lack individual data points and remain difficult to interpret. Because these problems directly affect the reliability of the reported results, I cannot confidently evaluate the Discussion or Conclusions until the statistical analysis is redone correctly and transparently reported. I encourage the authors to carefully review the analytical design, rerun the appropriate ANOVA models, and revise the manuscript accordingly before resubmission.
Response:
We sincerely thank the reviewer for the careful evaluation and constructive comments. We have thoroughly re-examined all statistical analyses and revised both the manuscript and figures accordingly to ensure complete transparency and correctness. These revisions ensure that the statistical methodology and results now correspond precisely to the experimental design and group structure. We believe the revised analyses and clarified reporting resolve the reviewer’s concerns regarding analytical validity and reproducibility.
The English writing has improved compared to the previous versions, with a clearer flow and fewer grammatical errors. However, many sentences remain overly long and complex, which affects readability. I recommend further language polishing by a fluent or professional editor to simplify sentence structure, ensure consistency in tense and terminology, and improve overall clarity.
Response:
We sincerely thank the reviewer for the insightful feedback regarding the linguistic quality and readability of our manuscript. In response, the entire text has been comprehensively revised to enhance clarity, streamline overly complex or lengthy sentences, and ensure consistent use of tense and terminology throughout. Furthermore, the manuscript has undergone additional professional English language editing by a fluent academic editor to ensure precision, fluency, and conformity with scientific writing standards. We believe that the revised version now demonstrates improved readability and stylistic consistency in line with the journal’s expectations.
Thank you for your valuable comments/suggestions and giving us the opportunity to revise the manuscript to a more readable level. We worked very hard to response your inquiries and to revise the manuscript. Before we finalized the revised manuscript, we have resent the manuscript for proof-reading as well as one last review by a native English-speaking researcher. We hope the manuscript could pass the review to be published in your prestigious journal: International Journal of Molecular Sciences
Sincerely yours,
Cheng-Chia Tsai
Department of Neurosurgery, Mackay Memorial Hospital, No. 92, Sec. 2, Zhongshan N. Rd., Taipei City 10449, Taiwan, ROC
Fax: +886-2- 2543-3642
Tel: +886-933888981
E-mail address: dschang580704@yahoo.com.tw (Cheng-Chia Tsai)